# Analysis of Migrating and Non-Migrating Tides of the Extended Unified Model in the Mesosphere and Lower Thermosphere

Matthew J. Griffith[1] and Nicholas J. Mitchell[2,3]

[1]Department of Mathematical Sciences, University of Bath, Claverton Down, Bath, BA2 7AY, UK.
[2]Department of Electronic & Electrical Engineering, University of Bath, Claverton Down, Bath, BA2 7AY, United Kingdom.
[3]British Antarctic Survey, High Cross, Madingley Rd, Cambridge, CB3 0ET, United Kingdom.

**Correspondence:** Matthew Griffith (M.J.Griffith@bath.ac.uk)

**Abstract.** Atmospheric tides play a key role in coupling the lower, middle and upper atmosphere/ionosphere. The tides reach large amplitudes in the Mesosphere and Lower Thermosphere (MLT) where they can have significant fluxes of energy and momentum and so strongly influence the coupling and dynamics. The tides must therefore be accurately represented in Global Circulation Models (GCMs) that seek to model the coupling of atmospheric layers and impacts on the ionosphere. The tides consist of both migrating (sun-following) and non-migrating (not sun-following) components, both of which have important influences on the atmosphere. The Extended Unified Model (ExUM) is a recently developed version of the Met Office's GCM (the Unified Model) which has been extended to include the MLT. Here, we present the first in-depth analysis of migrating and non-migrating components in the ExUM. We show that the ExUM produces both non-migrating and migrating tides in the MLT of significant amplitude across a rich spectrum of spatial and temporal components. The dominant non-migrating components in the MLT are found to be DE3, DW2 and DW3 in the diurnal tide and S0, SW1 and SW3 in the semidiurnal tide. These components in the model can have monthly mean amplitudes at a height of 95 km as large as 35 ms$^{-1}$/10 K. All the non-migrating components exhibit a strong seasonal variability in amplitude and significant short-term variability is evident. Both the migrating and non-migrating components exhibit notable variation with latitude. For example, the temperature and wind diurnal tides maximise at low latitudes and the semidiurnal tides include maxima at high latitudes. A comparison against published satellite and ground based observations shows generally good agreement in latitudinal tidal structure, with more differences in seasonal tidal structure. Our results demonstrate the capability of the ExUM for modelling atmospheric migrating and non-migrating tides and lay the foundation for its future development into a whole atmosphere model. To this end, we make specific recommendations on further developments which would improve the capability of the model.

## 1 Introduction

Atmospheric solar thermal tides are global-scale oscillations with a period exactly equal to one day or an integer fraction of one day. The solar thermal tides (hereafter, simply "tides") are excited primarily by the diurnal cycle in the solar heating of water vapour and ozone in the troposphere and stratosphere and the release of latent heat in deep tropospheric convection.

As the tides propagate upwards from their source regions their amplitudes increase because of the decreasing atmospheric gas density. In the mesosphere and lower thermosphere region (MLT) at heights of 80 - 100 km the tides cause large fluctuations in

winds, temperature, density and many other atmospheric parameters, including airglow emissions, ice-particle concentrations and trace-species densities. Tidal amplitudes in the MLT can exceed several 10s of m/s and they are often the largest-amplitude fluctuations of the MLT's field of waves.

Observations have revealed that the largest-amplitude tides in the MLT are the 24-hour diurnal and 12-hour semidiurnal tides. Generally, the semidiurnal tide is observed to reach maximum amplitudes at high latitudes near about 60°N/60°S but has small amplitudes at low latitudes, whereas the diurnal tide reaches maximum amplitudes at low latitudes but has much smaller amplitudes at middle and high latitudes (Mitchell et al., 2002; Davis et al., 2013; Mukhtarov et al., 2009; Pancheva et al., 2010).

The importance of tides lies in the key role they play in coupling the lower, middle and upper atmosphere/ionosphere (see reviews by Immel et al., 2006; Smith, 2012; Yiğit and Medvedev, 2015; Liu, 2016; Yiğit et al., 2016). For instance, the tidal winds modulate the fluxes of gravity waves (GWs) and so influence the wave forcing of the general circulation (e.g., Fritts and Alexander, 2003). The energy and momentum deposited by tides can cause a substantial warming of the MLT and a downward displacement of, and reduction in, the gravity-wave momentum transfer (wave drag) in the upper mesosphere (Becker, 2017). Tidal temperature fluctuations can cause variability in the occurrence of polar mesospheric clouds (Fiedler et al., 2005). The tides propagate upwards from the MLT into the thermosphere where they can modulate the ionospheric wind dynamo (e.g., Oberheide et al., 2009; Yiğit and Medvedev, 2015; Liu, 2016). The tides may also mediate the ionospheric response to sudden stratospheric warmings (e.g., Goncharenko et al., 2010).

An important distinction is between the migrating (sun-synchronous) tides and the non-migrating (not-sun-synchronous) tides. Here we will use the standard notation to identify the different tidal components. In this, a component is identified as either D or S to denote that it has diurnal or semidiurnal period; E or W to denote eastward or westward propagation and s = 0, 1, 2, 3... to denote its zonal wavenumber. A DW1 tide is thus a diurnal, westward propagating tide of wavenumber 1, an SE2 tide is a semidiurnal, eastward propagating tide of wavenumber 2 and a D0 or S0 is a standing diurnal or semidiurnal oscillation, respectively, with no zonal propagation or variation in phase (also known as a "breathing" component).

The migrating diurnal and semidiurnal components are thus the DW1 and SW2 components, respectively, that propagate westwards at sun-synchronous phase speeds and have zonal wavenumbers equal to the number of cycles of the tide per day. These tides are directly excited by the heating of the atmosphere by solar radiation. In contrast, the non-migrating tides are thought to be excited primarily by either i) longitudinal (land/sea) differences in the release of latent heat from deep tropospheric convection at tropical latitudes or ii) non-linear interactions between stationary planetary waves of zonal wavenumber 1 and the migrating tides. The latent-heat forcing is believed to primarily excite the diurnal components DE1, DE2, DE3, DW2, DW5, D0 and the semidiurnal components SW1, SE2, SW3 and SW6 (Forbes et al., 2003, 2007, 2008; Oberheide et al., 2006; Hagan and Forbes, 2002, 2003; Ekanayake et al., 1997; Oberheide et al., 2006). The non-linear interactions are thought to excite primarily the diurnal D0 and DW2 components and the SW1 and SW3 components (Hagan and Roble, 2001; Angelats i Coll and Forbes, 2002; Forbes and Wu, 2006; Murphy et al., 2009).

Tides propagating from the MLT into the thermosphere may drive significant modulation of F-region ionospheric density (see review by England, 2012). In general, although the migrating tides may produce strong day/night ionospheric variations, it

is the non-migrating tides that can produce longitudinal variations in the ionosphere. These latter tides can modulate F-region ionospheric density through mechanisms including i) electrodynamic coupling to the E-region dynamo, ii) plasma advection along geomagnetic field lines and iii) the modulation of photochemical equilibrium. Of particular note is that the conspicuous wavenumber-four structures observed in low-latitude total electron content are in part driven by a modulation of F-region density by a spectrum of non-migrating tidal components (particularly DE3) (Hagan et al., 2007; Forbes et al., 2008).

The important role of the tides in atmospheric coupling means that they must be represented accurately in models intending to span the lower, middle and upper atmosphere/ionosphere. However, it is recognised that there are major aspects of tides that remain challenging to model and that the causes of tidal variability remain uncertain (e.g., Smith et al., 2007; Baldwin et al., 2019). In particular, model biases remain in both the seasonal variability of tides and their short-term variability at time scales of less than a month (e.g., Dempsey et al., 2021; Chang et al., 2012; Hagan and Forbes, 2002; Oberheide et al., 2011; Ortland and Alexander, 2006).

Understanding the sources, propagation, variability and impacts of non-migrating tides is therefore crucial in attempts to investigate and model the coupling of atmospheric layers and the ionosphere. However, observational studies of non-migrating tides are limited by inherent difficulties in resolving the various migrating and non-migrating tidal components. For instance, there have been extensive ground-based observations made of tides in the MLT, in many cases made by meteor or MF radars (e.g., Murphy et al., 2007; Davis et al., 2013; Hibbins et al., 2019; Liu et al., 2020; Pancheva et al., 2021; Dempsey et al., 2021; Griffith et al., 2021). These radar observations usually offer excellent height and time resolution and are well suited to studies of tidal variability on time scales ranging from day-to-day to decadal - but observations made from a single site yield only the amplitudes, phases and vertical wavelengths of the superposition of migrating and non-migrating tides and cannot resolve the observed tidal oscillations into individual components.

In contrast, satellite instruments can make global observations, but are often limited by the need for the satellite to precess through local time in order to resolve the various non-migrating components. This limits the time resolution of the measurements such that, for instance, in many studies of non-migrating tides, TIMED/SABER measurements have an effective time resolution of about 60 days (e.g., Forbes et al., 2008) and UARS/HRDI and UARS/MLS have time resolutions of about 30 days (e.g., Forbes et al., 2003; Forbes and Wu, 2006).

These limitations in the ability of ground-based and satellite observations to resolve non-migrating tides mean that models must play an important role in efforts to understand their nature and variability.

"High-top" General Circulation Models (GCMs), which cover height ranges from the ground to the upper atmosphere, have considerable utility in the study of vertical coupling processes (e.g. Yiğit et al., 2016; Pogoreltsev et al., 2007; Akmaev, 2011). Such models play an important part in attempts to capture the variability of the thermosphere and ionosphere for space weather-forecasting, as well as in producing Whole Atmosphere Models (e.g., Jackson et al., 2019; Liu, 2016; Akmaev, 2011; Fritts et al., 2008).

A summary of several of the recent key non-mechanistic "high-top" GCMs is given in Griffith et al. (2021). Here, we simply note that a number of such models exist including the following, i) The Whole Atmosphere Model (WAM; Akmaev et al., 2008; Fuller-Rowell et al., 2008); ii) The Whole Atmosphere Community Climate Model with thermosphere and ionosphere

extension (WACCM-X; Liu et al., 2010, 2018); iii) The extended Canadian Middle Atmosphere Model (eCMAM; Beagley et al., 2000); iv) The Ground-to-topside model of the Atmosphere and Ionosphere for Aeronomy (GAIA; Fujiwara and Miyoshi, 2010; Jin et al., 2012, and references therein); v) The Hamburg Model of the Neutral and Ionized Atmosphere (HAMMONIA; Schmidt et al., 2006; Meraner and Schmidt, 2016); vi) The upper-atmosphere extension of ICON (Borchert et al., 2019); vii) The Entire Atmosphere GLobal model (EAGLE; Klimenko et al., 2019); viii) The HI Altitude Mechanistic General Circulation

Model (HIAMCM; Becker and Vadas, 2020); ix) The Coupled Middle Atmosphere Thermosphere-2 (CMAT-2; Yiğit et al., 2009); x) The University of Leipzig Middle and Upper Atmosphere Model (MUAM; Pogoreltsev, 2007; Pogoreltsev et al., 2007; Suvorova and Pogoreltsev, 2011); and xi) The whole atmosphere Kyushu GCM (Miyoshi and Fujiwara, 2008; Miyoshi and Yiğit, 2019).

    Several other models are also relevant in studies of tides and coupling. These include, i) The NCAR Thermosphere Iono-

sphere Mesosphere Electrodynamics General Circulation Model (TIME-GCM; Roble and Ridley, 1994; Hagan and Roble, 2001; Yamashita et al., 2010); ii) The linear mechanistic Global Scale Wave Model (GSWM; Hagan et al., 1999; Hagan and Forbes, 2002); and, iii) The Climatological Tidal Model of the Thermosphere (CTMT; Oberheide et al., 2011).

    In the context of these various "high-top" models, the new Extended Unified Model (ExUM; Griffith et al., 2020, 2021) extends the standard UM (Unified Model) (Walters et al., 2017) to the lower thermosphere. The model itself and its development

for the lower thermosphere is described further in Sect. 2.1, but we highlight here that the ExUM does not make the hydrostatic assumption and uses the deep-atmosphere equations of motion, making it a good candidate for modelling atmospheric tides.

    Griffith et al. (2021) investigated the ability of the ExUM to reproduce the observed winds and diurnal and semidiurnal tides of the MLT and compared them with meteor-radar observations at characteristic equatorial and polar locations (Ascension Island (8°S, 14°W) and Rothera (68°S, 68°W), respectively). The study demonstrated that, although there are biases in the

model tidal fields, they nevertheless capture many essential features of the observed tides. However, Griffith et al. (2021) did not decompose the model tidal fields into migrating and non-migrating components, nor did they examine the latitudinal structure of the tides beyond the two locations considered.

    It is also worth introducing here the importance of deposition of momentum by sub-grid scale non-orographic GWs, which must be accurately captured in parameterization schemes because of their important impact on tides in the MLT (e.g., Yiğit

and Medvedev, 2017; Yiğit et al., 2009; Miyahara and Forbes, 1991). For example, Yiğit and Medvedev (2017) provide an extensive discussion into the influence of parameterized small-scale GWs on the migrating diurnal tide. The gravity-wave scheme used in the ExUM is detailed in Sect. 2.

    Here we present the first use of the new ExUM to investigate the variability and latitudinal structure of tides in the MLT – the region where tidal amplitudes become large. We seek to answer the following scientific questions: i) what are the char-

acteristics of the combined migrating and non-migrating tidal components in the MLT of the new ExUM? ii) what is the contribution of individual migrating and non-migrating components at the high and low latitudes where the semidiurnal and diurnal components, respectively, are believed dominant? iii) how do the various tidal components in the ExUM compare with those observed? and iv) what improvements could be made in the ExUM to increase its ability to model tides in the MLT?

In Sect. 2 we describe the development of the ExUM version used. In Sect. 3 we present details of the principal non-migrating diurnal and semidiurnal tidal amplitudes and investigate the latitudinal and short-term variability of both the migrating and non-migrating tides[1]. As with Griffith et al. (2021), we use the characteristic equatorial and polar latitudes of Ascension Island (8°S) and Rothera (68°S). Finally, in Sect. 4 and Sect. 5 we place our results in the context of other tidal studies and consider how our results can guide future development of the ExUM.

## 2 Model development

### 2.1 The Extended Unified Model

The General Circulation Model (GCM) employed by the UK Met Office is the Unified Model (UM), which models both climate and weather forecast time scales with a unified approach. The model consists of two main parts – atmospheric dynamics and atmospheric physics. The former involves solving the Euler equations of motion governing atmospheric flow, and contains the *dynamical core* of the model; the latter attempts to make up for atmospheric physics not captured or resolved by the model dynamics, such as solar radiation and sub-grid scale GWs through *physical parameterizations* – see Walters et al. (2017) for more information on the complete formulation of the UM and Wood et al. (2014) for more information on the model dynamics.

The horizontal resolution is fixed at $1.25°$ N$\times 1.875°$ E and the vertical resolution is extended above the 85-level, 85 km standard UM configuration to a 100-level, 120 km configuration detailed below. Given the lack of modelled ionospheric effects such as ion drag in this model, we only consider fields up to around 110 km. This yields the previously mentioned Extended Unified Model which extends the working height of the standard UM into the lower thermosphere – initial work to perform this extension is discussed in Griffith et al. (2020). Following this research, the radiation scheme was extended to include non-LTE effects and the model temperature now contains the appropriate realistic forcing up to around 90 km. This work is detailed by Jackson et al. (2020) and discussed further in Griffith et al. (2021).

Latent heat release in the model is captured primarily through the UM convection schemes and associated large-scale cloud and cloud fraction schemes (see Sect. 2.5, 3.6.2 and 3.7 of Walters et al. (2017) for a more detailed description of the parameterizations used).

Other tidal dissipation processes such as eddy and molecular diffusion are not included in the MLT in this version of the ExUM[2]. Furthermore, the specific heats are not height varying, which is a reasonable assumption up to the turbopause which is the primary region of interest in this study.

The ExUM uses the non-orographic Ultra Simple Spectral Parameterization (USSP) of Warner and McIntyre (2001). The USSP scheme treats non-orographic GWs with non-zero phase speeds which are unable to be resolved by the model. The

---

[1]Note that the tidal phases are also an important consideration. However, these will not be presented here to keep the paper at a reasonable length. This is an interesting topic that will be addressed in future studies of the ExUM.

[2]Griffin and Thuburn (2018) suggest that molecular diffusion does not become dominant until around 150 km, however eddy diffusion is important in this region (e.g., Forbes and Hagan, 1988). In this initial stage of development of the model, these missing dissipative processes are primarily accounted for by increasing the vertical damping coefficient in the model (see Griffith et al. (2021) for more details), which is a proxy for these dissipative processes.

approach used is that of Warner and McIntyre (2001) with further modifications (Scaife et al., 2002) to launch an unsaturated spectrum from a level close to the surface and to impose a homogeneous (location invariant) total vertical flux of horizontal wave pseudomomentum. The spectrum uses a characteristic vertical wavelength peak of 4.3 km and parameterizes vertical wavelengths up to a maximum of 20 km. The amplitude of the spectrum is chosen to give momentum deposition and, hence, a Quasi-Biennial Oscillation (QBO) in the model that is realistic. For comparison with other parameterizations, a typical value of the total launch flux in all four directions is $6.6 \times 10^{-3}$ kgm$^{-1}$s$^{-1}$.

The inclusion of thermal effects is also important in the MLT (e.g., Yiğit and Medvedev, 2009; Medvedev and Klaassen, 2003; Hickey et al., 2011), and the USSP includes frictional heating due to gravity wave dissipation, and consequent loss of kinetic energy (see Walters et al. (2017) for more details), but does not include ionospheric heating effects such as ion drag. The aptitude of the USSP for use in the MLT and steps for its future development will be discussed in light of the results of this study.

Above around 90 km, the lack of appropriate high atmosphere chemistry and consequent heating via exothermic reactions means that the model temperature values cannot be assumed to be accurate. Given this lack of appropriate chemistry, a relaxation or nudging scheme to a climatological temperature field is used above 90 km (this scheme was first developed in Griffith et al. (2020) and more details can be found therein). Previously, as in Griffith et al. (2021), the temperature profile used in the nudging scheme was globally uniform, and so latitudinal variation in the MLT was only very weak, e.g. the summertime polar mesopause minimum was observed but not captured in a realistic manner. Thus, following this research, it was deemed that a more realistic temperature profile would be beneficial for the accuracy of the model in the MLT. To this end, the globally uniform temperature profile is replaced in this study with a temperature profile which varies by month and season, and with a varying mesopause height. This analytic temperature profile was calculated using a least-squares curve fitting algorithm, fitting to temperatures from the Committee on Space Research (COSPAR) International Reference Atmosphere (CIRA) (Fleming et al., 1990). Whilst this is an old data set, it gives a good climatological representation of atmospheric temperature up to 120 km. As well as this, the temperature profile produced for the nudging scheme only need provide an approximate representation of the atmospheric state.

To produce the analytic temperature profile $T_{\text{nudge}}(t, \phi, z)$ – a function of month ($t$), latitude ($\phi$) and height ($z$) – we first fit a function $T_{\min}$ of month ($t$) and latitude ($\phi$) to the minimum temperature value in the CIRA data found at the mesopause. The fit is of the form

$$T_{\min}(t, \phi) = a_{\text{T}} + b_{\text{T}} \cos\left(\frac{2\pi}{12} |t - 6|\right) \cos\left(\frac{\pi}{2} + \frac{2\pi}{360} \phi\right).$$

We then fit a function $z_{\min}$ of month ($t$) and latitude ($\phi$) to the height (height above sea level in metres) at which this mesopause temperature minimum occurs in the CIRA data. This results in an analytic profile for the height of the mesopause. The fit is of the form

$$z_{\min}(t, \phi) = a_{\text{z}} + b_{\text{z}} \cos\left(\frac{2\pi}{12} |t - 6|\right) \cos\left(\frac{\pi}{2} + \frac{2\pi}{360} \phi\right)$$

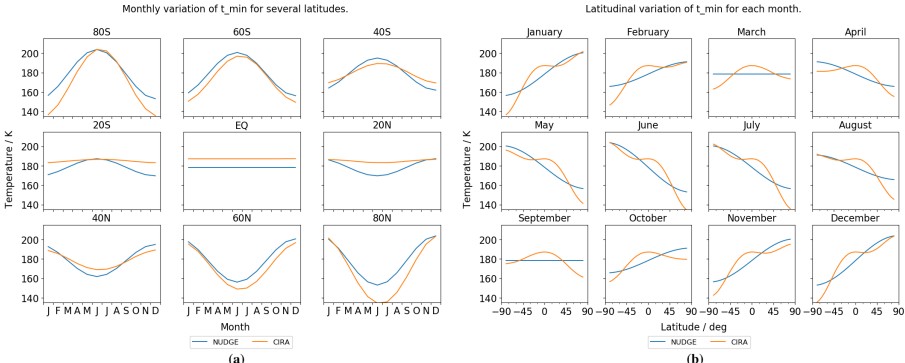

**Figure 1.** Variation of the fitted mesopause temperature profile $T_{\min}$ for **(a)** several latitudes as a function of month, and **(b)** all months as a function of latitude. The fitted function gives a reasonable fit for the purposes of the nudging scheme.

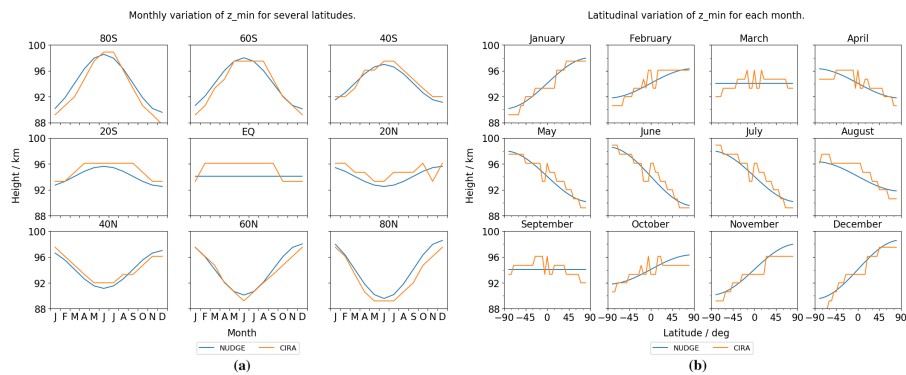

**Figure 2.** Variation of the fitted mesopause height profile $z_{\min}$ for **(a)** several latitudes as a function of month, and **(b)** all months as a function of latitude. The fitted function gives a reasonable fit for the purposes of the nudging scheme.

In summary, we now have an analytic expression for both the temperature at the mesopause and the height of the mesopause as a function of month and latitude. Fitting the parameters to the CIRA data yields $a_{\mathrm{T}} = 178.45$, $b_{\mathrm{T}} = 25.73$, $a_{\mathrm{z}} = 94065.91$ and $b_{\mathrm{z}} = 4561.23$. We compare the use of these analytic profiles with the CIRA data in Figures 1 and 2.

It can be seen that the analytic function gives a reasonable fit to the measured temperatures for the purposes of the nudging scheme – the analytic expression remains relatively simple and we avoid overfitting.

From this, the height dependence can be created. The temperature lapses linearly to the mesopause temperature minimum from below, and then a power law fit is used above the mesopause up to the current model lid at $120 \ \mathrm{km}$. Namely, at a height $z$ above the mesopause, we fit a function of the form

$$T_{\mathrm{nudge}}(t, \phi, z) = T_{\min}(t, \phi) + \Gamma_{\mathrm{thermo}} \left(z - z_{\min}(t, \phi)\right)^k.$$

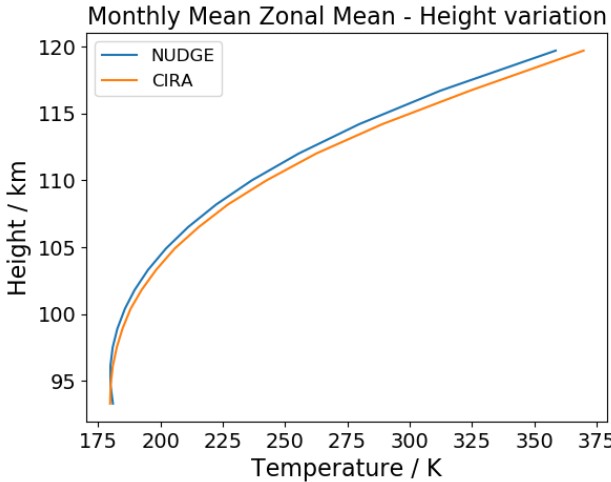

**Figure 3.** Variation of the zonal and monthly mean fitted nudging profile with height.

This fit yields parameters $\Gamma_{\text{thermo}} = 4.03 \times 10^{-9}$ and $k = 2.41$. The zonal and monthly mean variation in height above the mesopause can be seen in Figure 3. We observe a very good fit and the necessity of the power law fit is clearly demonstrated.

To summarise, this results in an ExUM which differs from the standard General Atmosphere (GA) 7.0 configuration of the UM (as described in Walters et al., 2017) in the following ways:

1. The model chemistry scheme is entirely switched off – the development of a chemistry scheme appropriate for the MLT is currently a work in progress.

2. Atmospheric aerosols are switched off and ozone background files are switched on.

3. The model upper boundary is raised from the standard $85 \, \text{km}$ to a height of $120 \, \text{km}$.

4. The forcing from the radiation scheme now includes non-LTE effects which means it is physically realistic up to $90 \, \text{km}$.

5. The temperature field above $90 \, \text{km}$ is nudged towards the prescribed monthly and latitudinally varying climatological temperature profile – this accounts for the lack of the chemistry scheme.

There will naturally be some variation in the modelled tidal fields when this background temperature profile is varied (e.g., Jones Jr. et al., 2018). However, the main focus of this work is to provide a closer look at the migrating and non-migrating components of atmospheric tides present in the newly extended model and show that they are of reasonable order of magnitude and compare reasonably with other models and with observations. A detailed analysis of the sensitivity of the tidal fields to the background temperature profile is beyond the scope of this work – we note that the goal in future development of the ExUM is to replace this background temperature profile with appropriate radiation and chemistry schemes for the MLT. As well as this, the primary diagnostics used are zonal and monthly mean fields for climatological variations, which will be less sensitive

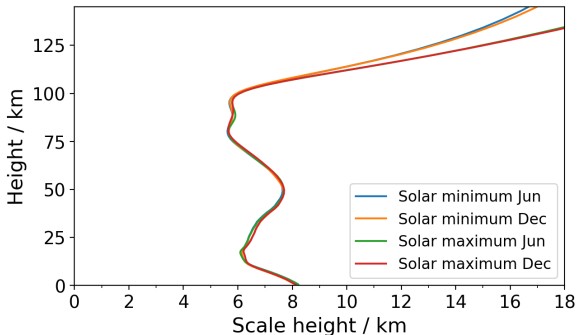

**Figure 4.** Atmospheric scale heights calculated using WACCM-X temperatures to give a baseline.

to such variations in the background temperature profile. Nevertheless, it is worth bearing this in mind when considering the results presented here.

We now describe the vertical level set used. The implementation builds on that used in Griffith et al. (2020) and Griffith et al. (2021). We move away from the fixed vertical level depth above the mesopause used previously, and instead use the atmospheric scale height to construct the vertical level set. This allows physically important vertical wave scales to be captured appropriately whilst relieving the numerical instabilities which can come from a fine vertical level set (e.g., Griffin and Thuburn, 2018; Griffith et al., 2020).

The implementation is as follows. The atmospheric scale height $H = RT/g$ is calculated for summer/winter conditions at both solar maximum and solar minimum using WACCM-X temperature values (e.g., Liu et al., 2010, 2018). This gives a reasonable baseline from which to calculate the vertical level set (see Figure 4). Naturally, the WACCM-X temperature profile and the CIRA climatological temperatures used for the background temperature profile will exhibit some differences, but both provide a reasonable initial implementation which can be tuned in future versions of the model

From this analysis, we decide to use zonal mean solar minimum conditions to create the vertical level set. This yields a vertical resolution which can capture wave scales appropriately throughout the solar cycle without the stringent condition imposed by using zonal minimum temperatures. With an upper boundary at $120$ km the effects of using the solar minimum temperature do not have much impact on the value of the scale height used, but with this condition in place the vertical level set can remain consistent when the upper boundary of the model is extended further into the thermosphere.

The vertical level depth remains the same as in the standard UM (namely increasing exponentially with increasing height from the lower boundary of the model), until the vertical depth reaches the value determined by the minimum value of $H/2$ found at the mesopause - we use $H/2$ to give a vertical 2 grid-point per scale height structure. At this point, we fix the vertical level depth at this value until the mesopause is reached.

Above the mesopause, the vertical level depth increases again with increasing height, and we use the value of $H/2$ to define each level depth. Namely, we add on a vertical level of depth $H/2$, read off the value of $H/2$ at the new atmospheric height

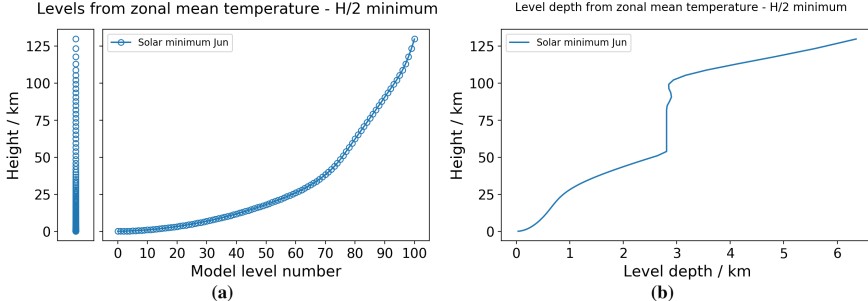

**Figure 5. (a)** Vertical level set and **(b)** corresponding level depths produced using the new implementation. The vertical level depth can be seen to be capped up to the mesopause, and then increase with increase in temperature going up through the thermosphere.

reached, and then add on a vertical level with this depth, and so on. Thus, the vertical level depths gradually become larger and larger as the model reaches higher into the thermosphere. The levels and vertical level depths produced by this method can be seen in Figure 5.

This completes specification of the model. The model runs are then all initialised using the same operational analysis from 1 September 2000 at 00 UTC. This allows the model to settle after the initialisation – known as the spin-up period of the model. Following this, climatological data is used to force background fields such as atmospheric ozone. Thus, we primarily examine climatological fields in this study – the main focus of this work is to provide a closer look at the migrating and non-migrating components of atmospheric tides present in the model.

An example of the climatological temperatures, zonal ($u$) winds and meridional ($v$) winds are provided for equinox and solstice conditions in Figure 6. The variation in the height of the mesopause can be clearly seen in the modelled temperature field. There are also still biases that exist in the model, such a summer wind reversal at middle latitudes which is at a lower altitude than expected and seasonal wind biases as discussed in Griffith et al. (2021). However, the goal of this paper is to provide an initial insight into the migrating and non-migrating tides present in the model and to educate improvements which can be made to correct these biases for future versions of the model.

The output attained from the model consists of hourly-sampled time profiles for temperature and both zonal and meridional wind fields for the whole of the model year considered – this high cadence is used so that diurnal and semidiurnal frequencies can be accurately resolved. For simplicity, we only show results for a single simulation, but multiple simulations were performed with exactly the same setup to ensure the robustness of the results, with no difference in model results observed between simulations. From these model fields, we compute several diagnostics to examine the properties of the tides produced by the model. We first extract the tidal perturbations by removing the mean from the model fields. We then decompose these tidal perturbations into diurnal and semidiurnal components in time, as well as several components in space. More precisely, we decompose the tidal perturbations by fitting a function of the form

$$F(t, \lambda) = A_0 + \sum_{i=1}^{2} \sum_{j=-6}^{6} A_{ij} \cos \left( it\frac{2\pi}{24} + j\lambda\frac{2\pi}{360} - \phi_{ij} \right),$$

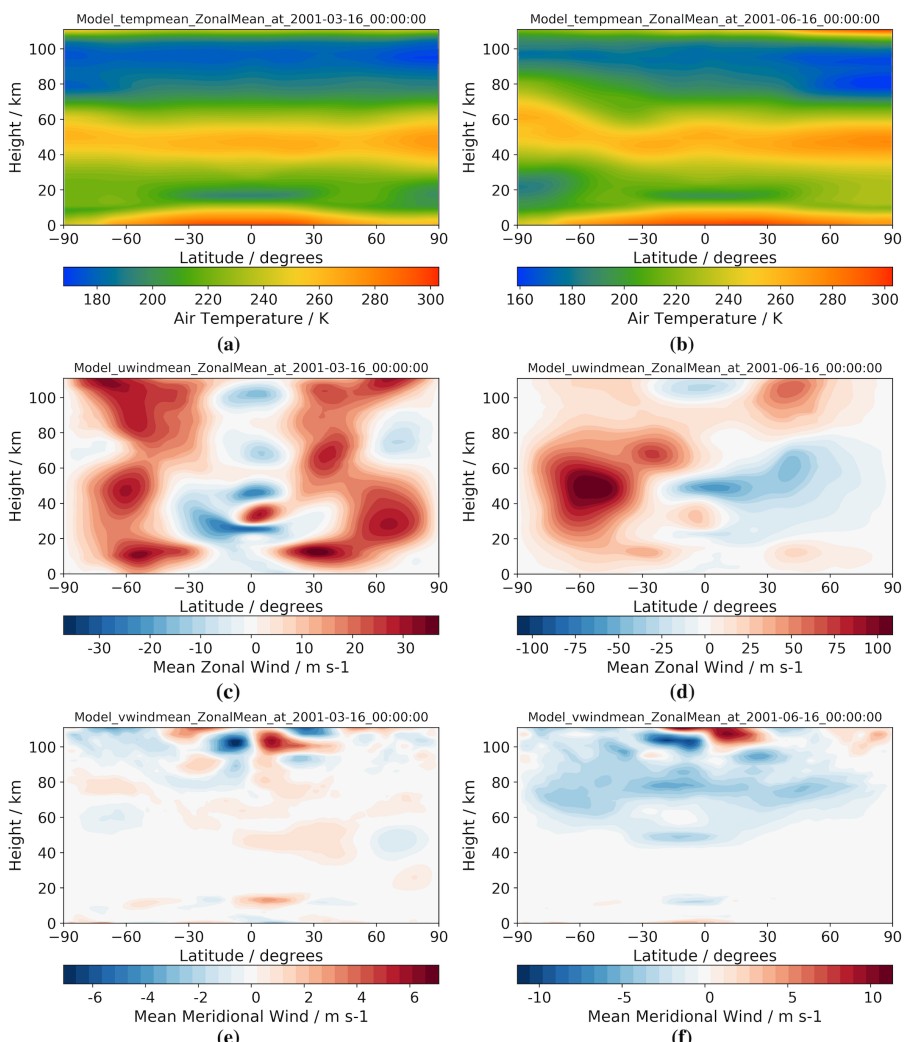

**Figure 6.** Latitude-height plot showing zonal and monthly mean fields for equinox (March) conditions for **(a)** temperature, **(c)** zonal ($u$) winds and **(e)** meridional ($v$) winds, and for solstice (June) conditions also for **(b)** temperature, **(d)** zonal ($u$) winds and **(f)** meridional ($v$) winds.

for a given model field $F$ varying in time (hours) and longitude (degrees). The amplitude of each component is then given by $A_{ij}$ with $\phi_{ij}$ the corresponding phase.

The temporal averaging of the tidal fitting is as follows. Where a figure shows tidal fields for a given month (e.g. Figure 12), the tidal fitting uses the values for the given month. Where a figure shows tidal fields for multiple months simultaneously (e.g. Figure 9), the tidal fitting uses a sliding 30-day window. The short term variation (Figure 14) uses tidal fitting on a 1-day sliding window.

## 3 Results

In this section, we present the ExUM migrating and non-migrating tides. We first look at instantaneous tidal perturbations as a function of latitude and height for the first day of January[3]. Here, we look at the total migrating and non-migrating components, without decomposition into separate spatial components. This provides some initial insight into the tidal properties of the modelled temperature, zonal and meridional wind fields as a superposition of all spatial components.

Following this, we restrict our attention to two latitudes, an equatorial latitude at $8°$S, and a polar latitude at $68°$S. We choose these latitudes to examine two key regimes, namely the equatorial regime, where the migrating diurnal tide is dominant; and the polar regime, where the migrating semidiurnal tide is dominant. Numerous observational studies have been performed at these latitudes (e.g. the studies performed using meteor radar at Ascension Island and Rothera by Davis et al. (2013) and Dempsey et al. (2021)) as well as the previous ExUM study by Griffith et al. (2021). For both these regimes, we first examine their variation with height using instantaneous tidal amplitudes as a function of longitude and height. Following this, we decompose the non-migrating portion of the tidal perturbations into its various spatial components using the fit described above on a 30-day sliding window. The plots for both the diurnal and semidiurnal temporal frequency and for the three model variables considered then highlights the variation in amplitude of each spatial component over the course of the year.

Having studied tidal properties at two latitudes, we then wish to examine the latitudinal properties of the modelled tides, to observe how amplitudes vary as a function of latitude. Again we decompose the tidal perturbations into their various spatial components and analyse how these vary as a function of latitude for both the diurnal and semidiurnal temporal frequencies and for the three model variables considered.

Finally, we return our attention to the equatorial and polar latitudes investigated previously to look at the short term variation in the tidal amplitudes of some of the dominant migrating and non-migrating components over the course of the year. We investigate this short term variability by calculating the amplitudes with a 24-hour sliding window and compare it to the standard 30-day sliding window used previously. This is to gain an insight into the "tidal weather" present in the model, which has been a recent topic of interest in the analysis of the MLT (e.g., Vitharana et al., 2019).

### 3.1 An initial exploration of model fields

We begin with an initial exploration of the model fields examined in this study – namely temperature, zonal ($u$) winds and meridional ($v$) winds. We fix a height of $95\,\mathrm{km}$ and plot instantaneous tidal perturbations from the modelled fields along with their decomposition into migrating and non-migrating components at 00UT on 1st January. These can be seen in Figure 7.

Of note is the size of the instantaneous tidal perturbations, which reach nearly 50 K in the modelled temperature field and around $140\,\mathrm{ms^{-1}}$ in the modelled winds.

The decomposition of these fields into migrating and non-migrating components reveals a migrating component that has a clear dominance in the DW1 component at equatorial latitudes, with a transition to a dominant SW2 component apparent on

---

[3]This is chosen solely for illustrative purposes, to give an initial insight into the tidal components present in the model, before showing more detailed tidal decompositions which are given for all months.

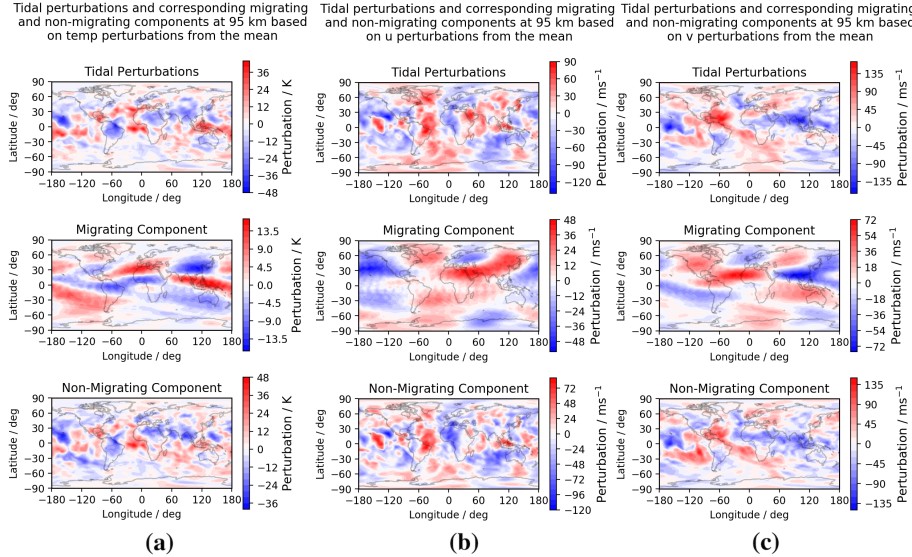

**Figure 7.** Longitude-latitude snapshot at 00UT on the first day of January of tidal perturbations at 95 km for **(a)** temperature, **(b)** zonal ($u$) winds and **(c)** meridional ($v$) winds. The equatorial DW1 tide and polar SW2 tide can be seen as the primary components of the migrating tide, with a superposition of several zonal wavenumbers apparent in the non-migrating components.

moving to polar latitudes. Namely, at equatorial latitudes one blue and one red region can be seen per latitude band, with a transition to two blue and two red regions at polar latitudes. The non-migrating component is of significant magnitude – up to nearly 50 K in temperature and 140 $\mathrm{ms}^{-1}$ in wind – and it is clear that it makes up a large portion of the tidal perturbation. The irregular nature of these fields indicate a superposition of several zonal wavenumbers and a need for further investigation – particularly given their large magnitude.

To this end, we examine the zonal wavenumber structure of the non-migrating tide in both an equatorial and polar regime in the following sections.

### 3.2 Equatorial regime

Firstly, we examine the height structure of the instantaneous tidal perturbations and corresponding migrating and non-migrating components of the model fields in the equatorial regime. Again we consider 00UT on 1st January. These can be seen in Figure 8.

Once more the amplitudes of the non-migrating component can be seen to contribute significantly to the overall tidal field – with magnitudes of up to 60 K in the temperature field and 170 $\mathrm{ms}^{-1}$ in the wind fields. Amplitudes of the tides can be seen to increase with increasing height which is consistent with the decrease in atmospheric density.

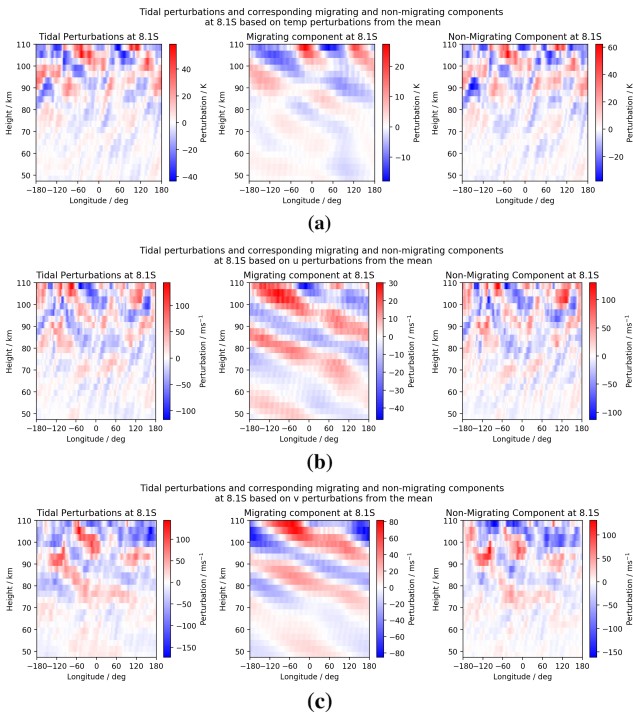

**Figure 8.** Longitude-height snapshot at 00UT on the first day of January at the equatorial latitude of Ascension Island ($8°$S) of tidal perturbations for **(a)** temperature, **(b)** zonal ($u$) winds and **(c)** meridional ($v$) winds. The equatorial DW1 tide can be seen as the primary component of the migrating tide, with some presence of the SW2 tide in temperature. A superposition of several zonal wavenumbers is apparent in the non-migrating components.

The migrating component of the temperature field appears to be dominated by the SW2 component above 60 km. In the wind fields, the migrating component is clearly dominated by the DW1 component at all heights. In all fields, the slope of the phase fronts is shallow indicative of a short vertical wavelength.

The non-migrating component is once more irregular but some structure can be seen, in particular a zonal wavenumber 3 structure around 90 km. In general the slope of the phase fronts appears to be steeper indicative of longer vertical wavelengths than those seen in the migrating component.

We now once more focus on a height of 95 km and decompose the non-migrating tidal field into its zonal wavenumber components using the method described in Sect. 2.1. With this we will be able to see which zonal wavenumbers are the dominant contributors to the non-migrating tide. We plot both diurnal and semidiurnal temporal frequencies in the equatorial regime for each zonal wavenumber across the year in Figure 9. We use a 30-day sliding average window centred on a given day.

The first feature of note is that the maximal amplitude of the diurnal tide is always larger than that of the semidiurnal tide in this equatorial regime. This is consistent with what is expected at an equatorial latitude where the diurnal tide should dominate.

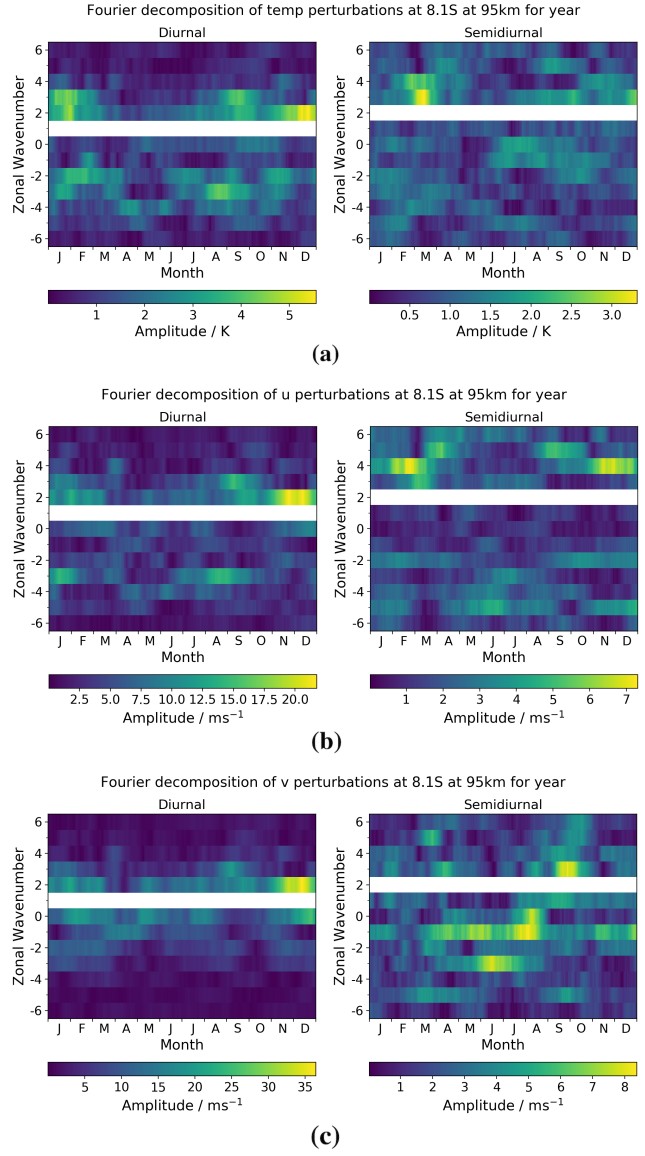

**Figure 9.** Diurnal and semidiurnal tidal amplitudes as a function of month and zonal wavenumber at the equatorial latitude of Ascension Island (8°S) for **(a)** temperature, **(b)** zonal ($u$) winds and **(c)** meridional ($v$) winds. The dominant migrating tidal component is removed in each case for clarity.

The magnitude of the semidiurnal tide in temperature is around 60% of that seen for the diurnal tide, which has a maximal amplitude of 5.5 K. In the zonal wind, the magnitude of the semidiurnal tide is around a third of that seen in the diurnal tide – which has a maximal amplitude of 22 ms$^{-1}$ – and in the meridional wind the semidiurnal tide is roughly a quarter of the observed diurnal tide, which has a maximal amplitude of 36 ms$^{-1}$.

We now focus on the modelled temperature field. In the diurnal component, we observe the largest non-migrating tidal amplitudes in the DW2 component, with a maximal peak of 5.5 K in December with amplitudes of 4 - 5 K also seen in January. Other non-migrating diurnal tidal amplitudes of note are DW3, which has maximal amplitudes of 4 - 5 K in January and September; DE2, which has maximal amplitudes of 4 - 5 K in January/February; and DE3 which has maximal amplitudes of 4 - 5 K in January and August. In the semidiurnal component, magnitudes are generally small, but peaks are seen in the

SW3 and SW4 tides, which have maximal amplitudes of around 3 K in March.

     Moving to the modelled zonal winds, in the diurnal component, the largest non-migrating tidal amplitudes are once more in the DW2 component. We observe a maximal peak of around $22 \, \mathrm{ms}^{-1}$ occurring in November/December. Other non-migrating diurnal components of notable magnitude are DW3, which peaks at around $15 \, \mathrm{ms}^{-1}$, and DE3 which peaks in January and August with a value of around $15 \, \mathrm{ms}^{-1}$. In the semidiurnal component, again magnitudes are small, but we observe maximal

amplitudes in the SW4 tidal component of around $7 \, \mathrm{ms}^{-1}$ in February/March and November/December. The SW5 component is also present, with maximal values of around $6 \, \mathrm{ms}^{-1}$ in March/April and September.

     Finally, we examine the modelled meridional winds. In the diurnal component maximal amplitudes of around $36 \, \mathrm{ms}^{-1}$ are seen in the DW2 component, occurring in November/December. Other tidal components of note are the "breathing" D0 component, which maximises with an amplitude of $25 - 30 \, \mathrm{ms}^{-1}$ in December; and the DW3 component where we see a peak

value of around $25 \, \mathrm{ms}^{-1}$ in September. In the semidiurnal component – which are of relatively small magnitude – we observe maximal amplitudes of around $8 \, \mathrm{ms}^{-1}$ spread across a number of components: SW3 which peaks in September/October; S0 which peaks in August; SE1 which sustains larger values from April through to August; and SE3 which peaks in June.

     In summary, the tidal properties in the equatorial tidal regime for i) modelled temperature, ii) modelled zonal wind and iii) modelled meridional wind are as follows:

– The instantaneous fields show maximal perturbation magnitudes at high altitudes of i) 60 K, ii) $140 \, \mathrm{ms}^{-1}$ and iii) $170 \, \mathrm{ms}^{-1}$.

     – The maximal amplitude of the diurnal non-migrating tidal components is always larger than that of the semidiurnal tide.

     – The DW2 component is the dominant diurnal non-migrating component across all fields, with maximal amplitudes of i) 5.5 K, ii) $22 \, \mathrm{ms}^{-1}$ and iii) $36 \, \mathrm{ms}^{-1}$. The DE3, DE2 and DW3 components are other components with notable

magnitudes.

     – The semidiurnal non-migrating components are small across the board, but relatively we see the largest magnitudes in i) SW3 & SW4, ii) SW4 & SW5 and iii) SE3, SE1, S0 & SW3.

     We perform a brief comparison with observations to place these results in the context of measured values. SABER values represent satellite measurements of temperature and TIDI & UARS values represent satellite measurements of wind – see Sect.

4 for more details. The magnitude of the migrating component is similar to observed values with some differences. Values of up to $22 \, \mathrm{ms}^{-1}$ are reported by Forbes et al. (2008) in SABER equatorial temperatures at $100 \, \mathrm{km}$, compared to ExUM values of around $15 \, \mathrm{ms}^{-1}$. In terms of wind, values of up to $40 \, \mathrm{ms}^{-1}$ and $70 \, \mathrm{ms}^{-1}$ are reported by Wu et al. (2008a) in TIDI equatorial

zonal and meridional winds (respectively) at 95 km, compared to ExUM values of around 30 ms$^{-1}$ and 70 ms$^{-1}$. A notable equatorial DW2 with smaller DE2 and DE3 components is also observed in TIDI equatorial zonal and meridional winds in

Oberheide et al. (2006), however they only observe maximal values of around 12 ms$^{-1}$ and 18 ms$^{-1}$ at 95 km compared to ExUM values of 22 ms$^{-1}$ and 36 ms$^{-1}$ for the zonal and meridional wind respectively. Finally, notable meridional equatorial SW4 zonal and SW3 meridional tidal components are also reported by Oberheide et al. (2007) in TIDI equatorial zonal and meridional winds at 95 km, however they also observe a notable SW1 meridional component which is not clear in the ExUM values. A notable SW3 meridional component is also reported by Angelats i Coll and Forbes (2002) in UARS equatorial

meridional winds.

Having examined the tidal spectrum in the equatorial regime, we now move on to the polar regime, where we expect the semidiurnal tide to dominate.

### 3.3 Polar regime

We now perform the same analysis in the polar regime. We again first examine the height structure of the instantaneous tidal

perturbations and corresponding migrating and non-migrating components of the model fields in this regime. We consider 00UT on 1st January. These can be seen in Figure 10.

The magnitude of the tidal perturbations is smaller in the polar regime than in the equatorial regime. It remains clear that the non-migrating component makes up a significant portion of the tidal field – up to almost 20 K in the temperature field and up to 120 ms$^{-1}$ in the wind fields. Again, the amplitudes of the tides increase with increasing height as the density decreases.

The migrating component of the temperature field is small, particularly when compared to the equatorial regime. It appears to be dominated by the DW1 component, less clearly so towards the top of the model where it is clear several components are superposed. In the instantaneous wind fields, there is a transition from a dominant DW1 component to a dominant SW2 component around 90 to 100 km. The slope of the phase fronts is steeper when compared with the equatorial regime, indicative of longer vertical wavelengths at this polar latitude.

The non-migrating component is again a superposition of many wavenumbers, but several finer wave structures can be seen. In particular around 90 - 95 km where we observe what appear to be zonal wave number 4 and 5 structures. There appear to be phase fronts indicating both westward and eastward propagation, as expected in non-migrating tides. The plots of the non-migrating component again highlight the need to decompose the field into its zonal wavenumber structure to provide a better picture on the zonal wavenumbers present in the model fields.

We now once more focus on a height of 95 km and decompose the non-migrating tidal field into its zonal wavenumber components using the method described in 2.1. We plot both diurnal and semidiurnal temporal frequencies in the equatorial regime for each zonal wavenumber across the year in Figure 9. We use a 30-day sliding average window centred on a given day.

We observe that, as expected, the maximal amplitude of the semidiurnal tide is always larger than that of the diurnal tide

in this polar regime. The magnitude of the diurnal tide in temperature is around 40% of that seen in the semidiurnal tide – it is worth noting that both have small magnitude however, with a maximal amplitude of around 1.6 K in the semidiurnal

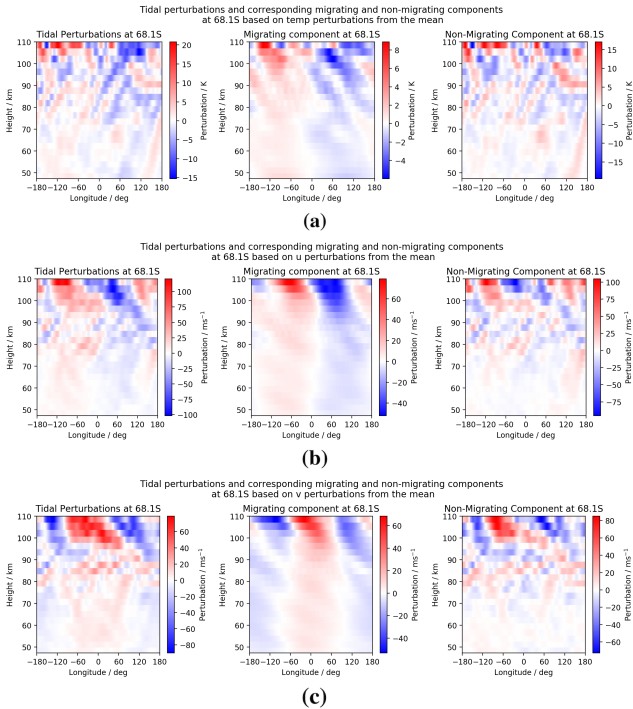

**Figure 10.** Longitude-height snapshot at 00UT on the first day of January at the polar latitude of Rothera (68°S) of tidal perturbations for **(a)** temperature, **(b)** zonal ($u$) winds and **(c)** meridional ($v$) winds. The equatorial DW1 tide can be seen as the primary component of the migrating tide at lower altitudes, with a switch to a dominant SW2 component occurring around 95 - 100 km in the wind fields and the temperature field becoming irregular. A superposition of several zonal wavenumbers is apparent in the non-migrating components.

component. The zonal wind has a diurnal component which is around 20% of the observed semidiurnal tidal amplitude, which maximises at around 14 ms$^{-1}$. Finally, the meridional wind has a diurnal component which is roughly 10% of the observed semidiurnal tidal amplitude, which maximises at around 13 ms$^{-1}$.

We comment first on the modelled temperature field. The magnitudes are small across both components, and therefore we will not place too much weight on observations made here. We see maximal amplitudes of around 1.6 K in the "breathing" S0 component in April/May and of around 0.6 K in the "breathing" D0 component in October.

The wind fields have larger magnitude. In the modelled zonal winds, we observe the largest non-migrating tidal amplitudes in the SW1 component, with a maximal value of around 14 ms$^{-1}$ occurring in August/September and with larger values of

around 10 ms$^{-1}$ also seen in May/June. Other notable non-migrating semidiurnal amplitudes are the "breathing" S0 component which peaks at around 8 ms$^{-1}$ in May and October. The diurnal component has small magnitude, and the largest values of around 2.5 ms$^{-1}$ are seen in the "breathing" D0 component in February and October.

Finally, we focus on the modelled meridional winds. The largest non-migrating tidal amplitude of around 13 ms$^{-1}$ is seen in the SW1 component in August/September, with large values of around 10 ms$^{-1}$ seen in June. There are once more some

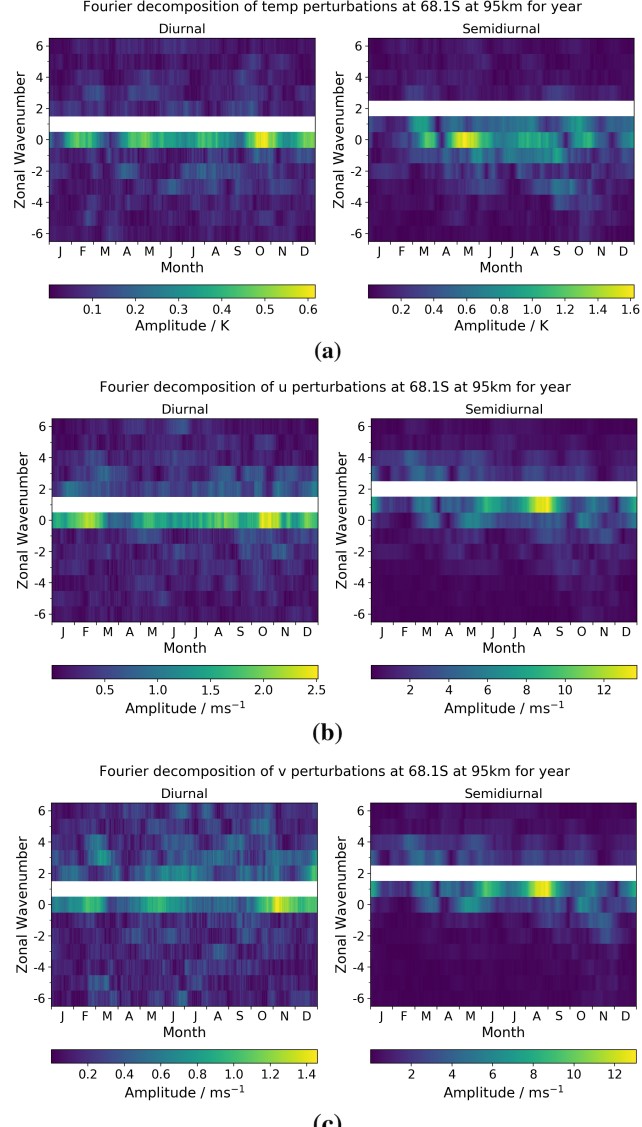

**Figure 11.** Diurnal and semidiurnal tidal amplitudes as a function of month and zonal wavenumber at the polar latitude of Rothera ($68°\mathrm{S}$) for **(a)** temperature, **(b)** zonal ($u$) winds and **(c)** meridional ($v$) winds. The dominant migrating tidal component is removed in each case for clarity.

larger values observed in the "breathing" S0 component also, with maximal values of around $8 \ \mathrm{ms}^{-1}$ occurring in May and October. Again, the diurnal component has small magnitudes, with maximal amplitudes of around $1.4 \ \mathrm{ms}^{-1}$ observed in the "breathing" D0 component in November.

In summary, the tidal properties in the polar tidal regime for i) modelled temperature, ii) modelled zonal wind and iii) modelled meridional wind are as follows:

– The instantaneous fields show maximal perturbation magnitudes at high altitudes of i) 20 K, ii) 120 ms$^{-1}$ and iii) 90 ms$^{-1}$.

     – The maximal amplitude of the semidiurnal non-migrating tidal components is always larger than that of the diurnal tide.

     – The SW1 component is the dominant semidiurnal non-migrating component across the wind fields, with the values in the temperature field being generally small. We observe maximal amplitudes of i) 1.0 K, ii) 14 ms$^{-1}$ and iii) 10 ms$^{-1}$.
The "breathing" S0 component also has notable magnitudes across all fields.

     – The diurnal non-migrating components are small across the board, but relatively we see the largest magnitudes in the D0 component in all fields.

We perform a brief comparison with observations to place these results in the context of measured values. The magnitude of the migrating component is similar to observed values. Values of 30 - 40 ms$^{-1}$ are reported by Angelats i Coll and Forbes
(2002) in UARS polar meridional winds at 95 km, compared to ExUM values here of around 30 ms$^{-1}$. A notable SW1 polar meridional tidal component is also reported by Angelats i Coll and Forbes (2002) in UARS winds. However, the values observed at 95 km are closer to 4 ms$^{-1}$ and values closer to 10 ms$^{-1}$ (as seen in the ExUM at 95 km) are only observed at 105 - 110 km. However, the polar SW1 component of TIDI polar zonal and meridional winds reported in Wu et al. (2011) at 95 km are up to 12 ms$^{-1}$ in both the zonal and meridional components, in keeping with the values seen in the ExUM. Finally, the
non-migrating components of the TIDI polar zonal winds reported in Wu et al. (2008b) show notable DE3, DE2, DE1, D0 and DW2 magnitudes (around 12 ms$^{-1}$) which we do not observe in the ExUM at polar latitudes. The reason for this difference is at this stage unknown.

We now have a good grasp of the dominant non-migrating tidal components in two key regimes – at an equatorial and polar latitude. We now wish to get a better understanding of how the components of the tide vary with latitude, and so we examine
this in the following section.

### 3.4   Latitudinal dependence

Here, we extract the latitudinal dependence of the tides, by examining the amplitudes of the spatial components as a function of latitude for each month of the year. We include the migrating component in this analysis, and remove zonal wavenumber 5 and 6 – which are generally small – to help with visualisation. In Figure 12 we plot the diurnal tidal amplitudes for the spatial
components considered for each month. In Figure 13 we repeat the analysis but for the semidiurnal tidal amplitudes.

We first turn our attention to the modelled temperature field. We observe maximal tidal amplitudes of around 16 K. Looking at the migrating (DW1) component, we see a clear three-peak structure, with the largest peak observed at the equator and the two smaller peaks at latitudes of approximately 30°S and 30°N. We observe maximal amplitudes in June and November with a pronounced minimum in August. Looking at the non-migrating components, we observe that the DW2 component is by far
the largest with amplitudes of up to around 11 K at the equator in December where it is nearly as large as the diurnal migrating component. It also has large amplitudes in November of 7 - 8 K and in May and September when it reaches around 5 K at the

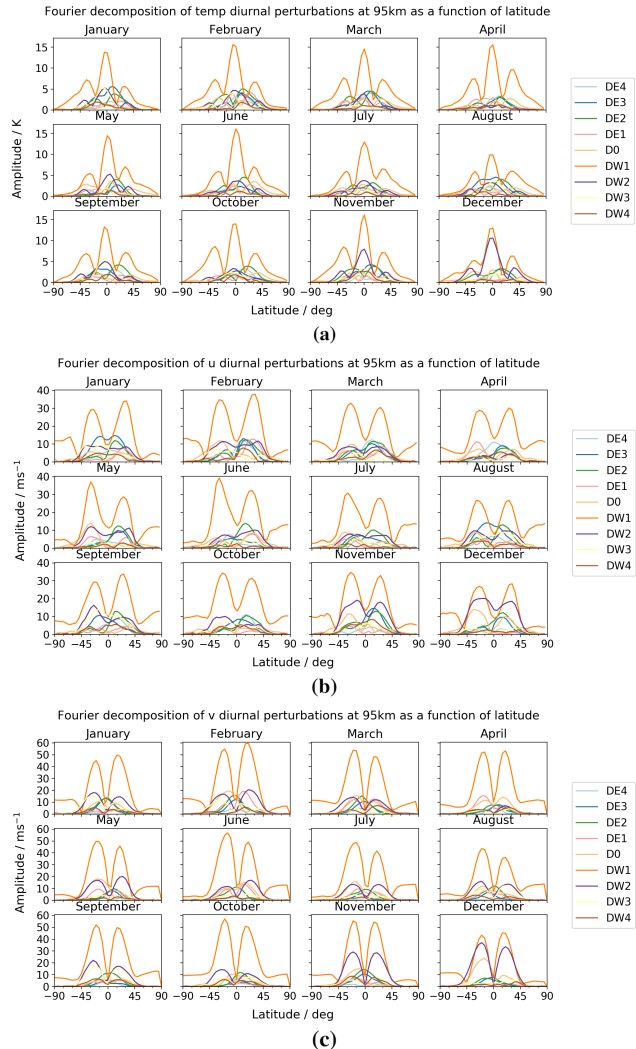

**Figure 12.** Latitude-amplitude plot of diurnal tidal amplitudes across the year for **(a)** temperature, **(b)** zonal ($u$) winds and **(c)** meridional ($v$) winds.

equator. We also observe that it has a similar three peak structure. Other large components of note are DE3, DE2 and DW3. DE3 generally has a one peak structure in a 20°S to 20°N band around the equator which reaches a maximal amplitude of around 5 K in January and August. DE2 generally has a two peak structure with these peaks occurring at the minima of the DW1 component at around 25°S and 25°N, and with maximum amplitudes of around 5 K in February and June. Finally, the DW3 component generally has a three peak structure in line with the structure observed in the migrating component. We see maximal amplitudes of this component of around 5 K in January and September.

We focus now on the modelled zonal winds, where we observe maximal tidal amplitudes of around 40 ms$^{-1}$. In the migrating (DW1) component, we see a clear two peak structure with large peaks at approximately 20 - 30°S and 25 - 30°N


with a minimum at the equator. Some increase towards the south pole is evident in the austral spring/summer period (October, November, December, January, February). Maximal amplitudes occur in February, June and October/November, whilst minimal amplitudes occur in April and August. Turning our attention to the non-migrating components, we once more observe a dominant DW2 component with amplitudes up to around $20\,\mathrm{ms^{-1}}$ in December matching that of the diurnal migrating tide. In general it also has the same two peak structure as the migrating component. DW2 is large in November also reaching around

$20\,\mathrm{ms^{-1}}$, and in May and September where it reaches $10$ - $15\,\mathrm{ms^{-1}}$. Many other non-migrating components also have large amplitudes in different months of the year. DE4 maximises at $12\,\mathrm{ms^{-1}}$ at around $15°$N in March/April and October. DE3 maximises at around $17\,\mathrm{ms^{-1}}$ in the region of $15°$S to $15°$N in January and November. DE2 reaches values of around $15\,\mathrm{ms^{-1}}$ at $15°$N for a large part of the year. DE1 maximises at around $12\,\mathrm{ms^{-1}}$ at $30°$S in April and at $30°$N in January. The "breathing" D0 component reaches a value of $15\,\mathrm{ms^{-1}}$ at around $30°$S in February, May and December. Finally, the DW3 component

maximises at around $15\,\mathrm{ms^{-1}}$ at around $15°$S in September.

Finally we look at the modelled meridional winds. These have the largest maximal amplitudes seen so far of around $60\,\mathrm{ms^{-1}}$. The migrating (DW1) component shows a similar two peak structure to that observed in the zonal wind, with the peaks similarly located around $20°$S and $20°$N with a pronounced minimum at the equator. Again, some increase is seen towards the south pole in the austral spring/summer period but it is relatively less pronounced when compared to the zonal wind. The maximal

amplitudes also follow the same monthly pattern as the zonal winds; we see maxima in February, June and October and minima in Apr/May and August. Looking at the non-migrating components, DW2 is again dominant, follows a two peak structure and has maximal amplitude in December of around $40\,\mathrm{ms^{-1}}$ comparable with the amplitude of the diurnal migrating component. DW2 is also large in November with a maximal amplitude around $30\,\mathrm{ms^{-1}}$, and through much of the rest of the year with amplitudes near $20\,\mathrm{ms^{-1}}$ (it is at its smallest in April with amplitudes below $10\,\mathrm{ms^{-1}}$). Many other non-migrating components

are also large as was observed with the zonal winds. DE3 generally has a one peak structure maximising at the equator with values around $15\,\mathrm{ms^{-1}}$ in January. DE2 has a similar one peak structure with maximal values of $15\,\mathrm{ms^{-1}}$ at the equator in January, February, March and November. DE1 reaches values of $16\,\mathrm{ms^{-1}}$ at $15°$S in April and $20\,\mathrm{ms^{-1}}$ at $15°$N in February. The "breathing" D0 component maximises at $15°$S with a value of $19\,\mathrm{ms^{-1}}$ in February and with a value of $22\,\mathrm{ms^{-1}}$ in December. Finally, DW3 generally has a two peak structure maximising around $20°$S and $20°$N with values of $15\,\mathrm{ms^{-1}}$ in

January and August and $18\,\mathrm{ms^{-1}}$ in September.

Having performed an in depth analysis of the diurnal tidal components, we now look at the variation of the semidiurnal tidal components with latitude, presented in Figure 13.

We first focus on the modelled temperature field, where we see maximal tidal amplitudes of around 11 K which are less than those seen in the diurnal migrating component. In the migrating (SW2) component, we generally observe a three peak

structure - occasionally one of the peaks breaks down leaving a two peak structure remaining. The central peak generally occurs between $10°$S and $10°$N with the left and right peaks occurring approximately $30°$ north or south of the central peak. We observe maximal amplitudes in May/June/July and minimal amplitudes in October/November/December. Turning our attention to the non-migrating semidiurnal tidal components, there is no clear largest component. The SE2, SW1, SW3 and SW4 components represent the largest of the non-migrating semidiurnal components. SE2 has maximal amplitudes of 3 - 4 K

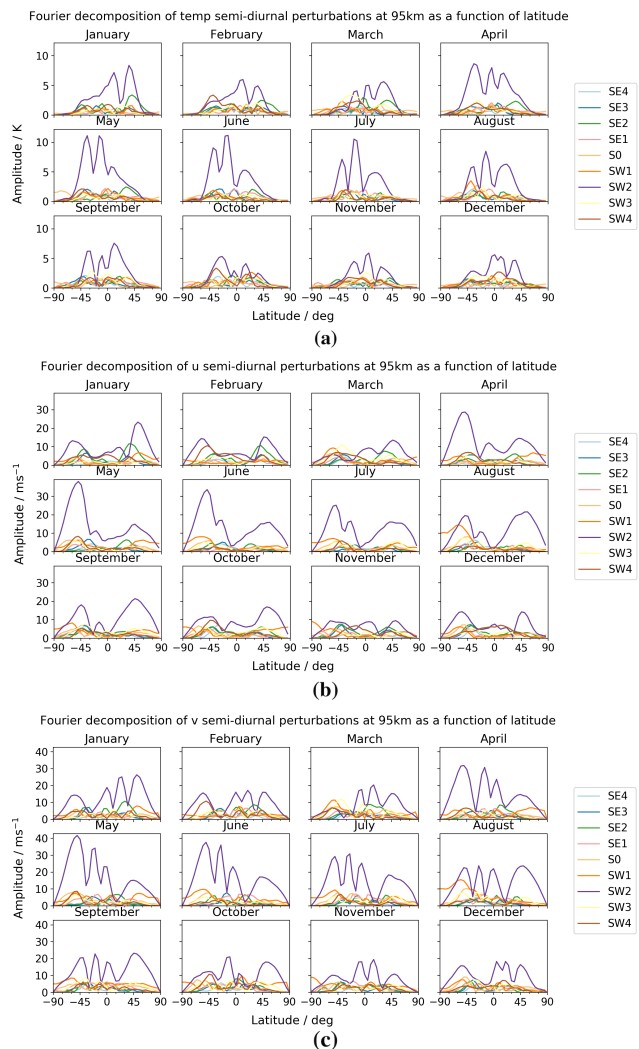

**Figure 13.** Latitude-amplitude plot of semidiurnal tidal amplitudes across the year for **(a)** temperature, **(b)** zonal ($u$) winds and **(c)** meridional ($v$) winds.

around $40°$N for most of the first half of the year. Peak amplitudes of around 3 - 4 K are also seen for the SW1 component at $40°$S in August; and for the SW3 component at $20°$S in March. Finally, the SW4 component reaches values of 4 K at $40°$S in February and at around $30°$S in October.

We now look at the modelled zonal winds. We observe maximal tidal amplitudes of around 40 $\mathrm{ms}^{-1}$, which are similar to those seen in the diurnal migrating component. Looking at the migrating (SW2) component, we generally see a two peak structure, but a third smaller peak often occurs between these peaks. Generally, the two largest peaks occur at approximately $50°$S and $50°$N and there is often a third peak between these occurring anywhere between $30°$S and $30°$N. Maximal amplitudes are seen in May/June, with minimal amplitudes in November/December. In general the peak amplitude at $50°$S is greater than

or equal to the peak amplitude observed at 50°N. Now looking at the non-migrating components, again there is no outright largest non-migrating tide. As in the temperature field, SE2, SW1, SW3 and SW4 have the largest amplitudes. SE2 tends to have a two peak structure with maximal values at 40°S and 40°N. It maximises with values of $10\ \mathrm{ms}^{-1}$ at these latitudes in January/February. The SW1 component tends to peak towards the south pole. It has maximal amplitudes at 60°S, with a value of $10\ \mathrm{ms}^{-1}$ in March and $15\ \mathrm{ms}^{-1}$ in August. SW3 reaches a peak value of around $10\ \mathrm{ms}^{-1}$ at around 40°S in March. Finally, the SW4 component maximises with a value of $10\ \mathrm{ms}^{-1}$ at 50°S in February and at 40°S in October.

Finally, we analyse the modelled meridional winds. We observe maximal tidal amplitudes similar to those seen in the zonal winds of around $40\ \mathrm{ms}^{-1}$ making them smaller than those seen in the diurnal migrating component. The migrating (SW2) component generally has a four peak structure with the two outer peaks centred around approximately 50°S and 50°N and the two central peaks moving to the north and south of the equator about 25 degrees apart. The tide has maximal amplitudes around May/June and has minimal amplitudes in November/December. Looking at the non-migrating components, again there is no clear dominant component and SE2, SW1, SW3 and SW4 all have notable magnitudes. SE2 component maximises at 30°N in June with a value of $10\ \mathrm{ms}^{-1}$. Similar to the zonal wind SW1 component, the SW1 component here also has its largest amplitudes towards the south pole. We observe maximal amplitudes of $18\ \mathrm{ms}^{-1}$ at around 55°S in August. The SW3 component peaks at $13\ \mathrm{ms}^{-1}$ in March at 40°S. Finally the SW4 component has a maximal amplitude of $13\ \mathrm{ms}^{-1}$ seen in February at 50°S, and values of $10\ \mathrm{ms}^{-1}$ seen at 50°S in May and at 45°S in October.

In summary, the tidal properties as a function of latitude for i) modelled temperature, ii) modelled zonal wind and iii) modelled meridional wind are as follows:

- Maximal diurnal tidal amplitudes are i) 16 K, ii) $40\ \mathrm{ms}^{-1}$ and iii) $60\ \mathrm{ms}^{-1}$, which are produced by the migrating (DW1) component.

- The diurnal migrating component has a i) three peak, ii) & iii) two peak structure.

- The dominant diurnal non-migrating component is the DW2 component across all fields with maximal amplitudes of i) 11 K, ii) $20\ \mathrm{ms}^{-1}$ and iii) $40\ \mathrm{ms}^{-1}$. Other components of notable magnitude are the i) DE3, DE2 & DW3, ii) & iii) DE3, DE2, DE1, D0 & DW3.

- Maximal semidiurnal tidal amplitudes are i) 11 K, ii) $40\ \mathrm{ms}^{-1}$ and iii) $40\ \mathrm{ms}^{-1}$, which are produced by the migrating (SW2) component.

- In general, the semidiurnal migrating component has a i) three peak, ii) two peak and iii) four peak structure.

- The dominant non-migrating semidiurnal components are the SE2, SW1, SW3 and SW4 components across all fields, with maximal amplitudes of i) 4 K, ii) $15\ \mathrm{ms}^{-1}$ and iii) $18\ \mathrm{ms}^{-1}$.

We have now detailed the variation in diurnal and semidiurnal tidal amplitudes with latitude for the various spatial components considered. It is now worthwhile to consider variation on a finer time scale – namely short term variability – which we focus on for the final section of our analysis.

 **3.5 Short term variability**

Here we perform an analysis of the short term variability present in the amplitude of the tidal components. This is primarily to investigate the magnitude of such perturbations. To do this, we apply the analysis to a 30-day sliding window, and contrast it with that from a 1-day sliding window. In Figure 14, we present the variability in the migrating and some of the larger non-migrating tidal components across the course of the year within the two regimes we considered previously – namely an equatorial and a polar latitude. The bold line represents the value from the 30-day sliding window, and the faded line represents the value obtained using the 1-day sliding window.

Note that when we discuss the short term variations below, these are given *relative to the value for the 30-day sliding window*. Namely, we discuss the percentage difference between the faded line and the solid line in Figure 14.

We first analyse the modelled temperature field. Looking at the migrating components (DW1 and SW2) in the equatorial regime, we see maximal amplitudes of around 15 K. We observe a peak in DW1 amplitudes in January/February of around 8 K with short term variation of up to 4 K throughout the year (i.e. at least a 50% variation). The SW2 component here peaks at a maximal value of around 12 K in May/June with short term variation of up to 3 K throughout the year (i.e. at least a 25% variation). In general, SW2 has larger amplitudes in April to September (equatorial Spring/Summer) with smaller amplitudes in October to March (equatorial Autumn/Winter). The migrating components in the polar regime are relatively small throughout the year for both components with little short term variation. We focus on a subset of the non-migrating components which have larger magnitudes within each of the two regimes, with peak values of around 12 K. In the equatorial regime, we focus on the DE3, DW2 and DW3 tidal components. DE3 peaks in January and August with values around 4 K, with short term variation up to 5 K throughout the year (i.e. short term variation of 125%). DW2 has maximal values in January and December of around 5 K with short term variation up to to 5 K throughout the year (i.e. short term variation of around 100%). Finally, DW3 peaks in January/February and September with values around 5 K and with short term variation of up to 7 K – the largest short term variation seen of 140%. The non-migrating components in the polar regime are also relatively small – perhaps the only point of note is the short term variation in the "breathing" S0 component which varies by up to 1 K, or around a 75% variation.

We now turn our attention to the modelled zonal winds. First focusing on the migrating components (DW1 and SW2) in the equatorial regime, we observe maximal amplitudes of around 35 $\text{ms}^{-1}$. Looking at the DW1 component, we see peak amplitudes of around 30 $\text{ms}^{-1}$ in March/April and October/November, with short term variation of up to 12 $\text{ms}^{-1}$, or a 40% variation. The SW2 component peaks in April/May with amplitudes of around 12 $\text{ms}^{-1}$ with short term variation of up to 5 $\text{ms}^{-1}$, or around a 40% variation. In the polar regime, we observe larger amplitudes than those seen in the temperature field. The dominant SW2 component peaks in April/May with a value of 28 $\text{ms}^{-1}$, with short term variation of up to 10 $\text{ms}^{-1}$, or around a 35% variation. It is notable that this large peak amplitude follows near zero amplitude values in the preceding month. The DW1 component in the polar regime has maximal values of 12 $\text{ms}^{-1}$ in January and December with short term variation of around 3 $\text{ms}^{-1}$, or a 25% variation. The amplitudes seem to experience a six month low in April through September, following by a six month high from October through March which is also observed to a lesser extent in the temperature field. Moving to

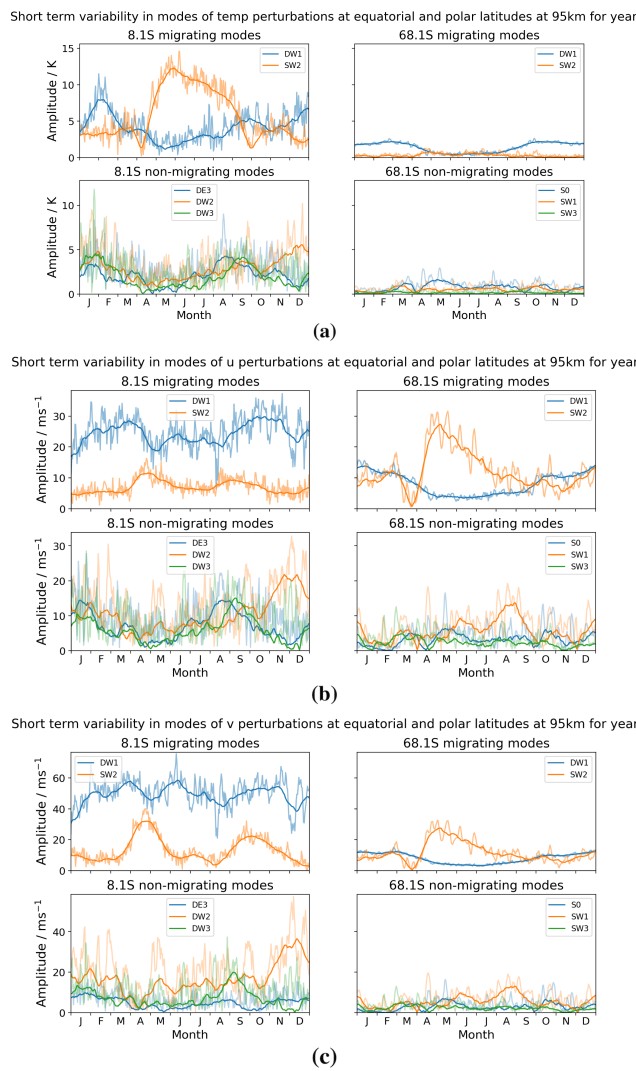

**Figure 14.** Tidal amplitudes as a function of time for the latitudes of $8°$S and $68°$S showing the short term variability of the migrating and largest non-migrating tidal components over the course of the year for **(a)** temperature, **(b)** zonal ($u$) winds and **(c)** meridional ($v$) winds. The bold line represents the value from the 30-day sliding window, and the faded line represents the value obtained using the 1-day sliding window.

the non-migrating components, we observe maximal amplitudes similar to those seen in the migrating components of around 35 ms$^{-1}$. We again focus on the DE3, DW2 and DW3 components in the equatorial regime. DE3 peaks in January and August with values around 14 ms$^{-1}$, with short term variation of up to 12 ms$^{-1}$, or around a 85% variation. These peaks line up with the peaks in the temperature field seen previously. DW2 has maximal values of 20 ms$^{-1}$ observed in November/December with large short term variation of up to 20 ms$^{-1}$, or a 100% variation. Finally, DW3 peaks in January and September with values of 10 and 14 ms$^{-1}$ respectively. The short term variation seen here is some of the largest seen in the zonal winds, with variation up

to 18 - 19 $\text{ms}^{-1}$, or around a 130% variation. Finally, we look at the non-migrating component in the polar regime, and focus on the S0, SW1 and SW3 components. The "breathing" S0 component peaks with values of 6 $\text{ms}^{-1}$ in May with large short term variation of up to 10 $\text{ms}^{-1}$ or around 165%. The SW1 component has maximal values in August/September of around 12 $\text{ms}^{-1}$ with large short term variation of up to 15 $\text{ms}^{-1}$ or 125%. The SW3 component peaks around February/March/April with values of around 5 $\text{ms}^{-1}$, and again with large short term variation of around 7 - 8 $\text{ms}^{-1}$, or around 150%.

Finally we come to the modelled meridional winds. In the migrating components (DW1 and SW2) we observe a very similar pattern to the migrating components seen in the zonal wind field, with similar amplitudes in the polar regime, but with almost double the amplitude in the equatorial regime, giving maximal amplitudes of around 70 $\text{ms}^{-1}$. The DW1 component in the equatorial regime has the same March/April and October/November peak seen in the zonal winds, with amplitudes here of around 60 and 55 $\text{ms}^{-1}$ respectively. In the meridional winds we also see larger values in June of near 60 $\text{ms}^{-1}$. The short term variation seen is up to 20 $\text{ms}^{-1}$, or around 33% of the base value. The SW2 component has more pronounced peaks in April/May and September/October (the equinoxes) than the zonal winds, with values of 30 $\text{ms}^{-1}$ and 20 $\text{ms}^{-1}$ respectively. Short term variation occurs up to a value of 10 $\text{ms}^{-1}$, or around a 33 - 50% variation. Moving on to the migrating components in the polar regime, as noted previously these have very similar structure and magnitude to the migrating components seen in the zonal winds and so we refer the reader to this analysis. The non-migrating components see maximal amplitudes of around 60 $\text{ms}^{-1}$, which is similar to the maximal amplitude seen in the migrating components. In the equatorial regime we again focus on the DE3, DW2 and DW3 components. DE3 component has consistently smaller amplitudes than those seen in the corresponding component in the zonal winds, with amplitudes always less than 10 $\text{ms}^{-1}$. The short term variation is still pronounced however with a magnitude of up to 10 $\text{ms}^{-1}$, or over a 100% variation. The DW2 component has a very similar structure to that seen in the zonal winds, but with almost double the magnitude, peaking in November/December with a value of 38 $\text{ms}^{-1}$. We observe short term variation of up to 22 $\text{ms}^{-1}$, or a variation of nearly 60%. Finally, coming to the DW3 component, we see a similar structure to that seen in the zonal wind, but with a larger peak in August/September of around 20 $\text{ms}^{-1}$, and a slightly larger peak in January/February of around 14 $\text{ms}^{-1}$. Short term variation seen here is at most 15 - 20 $\text{ms}^{-1}$, or around a 100% variation in general. We now approach the non-migrating tidal components in the polar regime, and again focus on the S0, SW1 and SW3 components. As with the migrating components in the polar regime, these have very similar structure and magnitude to that seen in the zonal wind non-migrating components and so we refer the reader to this analysis.

In summary, the tidal properties considering short term variability for i) modelled temperature, ii) modelled zonal wind and iii) modelled meridional wind are as follows:

- Maximal amplitudes of the migrating components in the equatorial regime are, for DW1, i) 8 K, ii) 30 $\text{ms}^{-1}$ and iii) 60 $\text{ms}^{-1}$, and for SW2, i) 12 K, ii) 12 $\text{ms}^{-1}$ and iii) 30 $\text{ms}^{-1}$.

- Maximal amplitudes of the migrating components in the polar regime are, for DW1, i) <5 K, ii) 12 $\text{ms}^{-1}$ and iii) 12 $\text{ms}^{-1}$, and for SW2, i) <5 K, ii) 28 $\text{ms}^{-1}$ and iii) 28 $\text{ms}^{-1}$.

- Short term variation or "tidal weather" in the migrating components can lead to a percentage variation (relative to the 30-day sliding window values) of up to i) 50%, ii) 40% and iii) 50%.

- Maximal amplitudes of the non-migrating components considered in the equatorial regime are i) 5 K, ii) 20 $ms^{-1}$ and iii) 38 $ms^{-1}$.

- Maximal amplitudes of the non-migrating components considered in the polar regime are i) <5 K, ii) 12 $ms^{-1}$ and iii) 12 $ms^{-1}$.

- Short term variation or "tidal weather" in the diurnal non-migrating components considered can lead to a percentage variation (relative to the 30-day sliding window values) of up to i) 140%, ii) 130% and iii) 100%.

- Short term variation or "tidal weather" in the semidiurnal non-migrating components considered can lead to a percentage variation (relative to the 30-day sliding window values) of up to i) 75%, ii) 165% and iii) 165%.

This completes our analysis of the migrating and non-migrating tidal components observed in the modelled temperature, zonal and meridional wind fields from the Extended Unified Model, and we proceed to put these results in the context of other modelling and observational studies in the discussion which follows.

## 4   Discussion

In the results presented above, we observe significant magnitude and structure in the components of both the migrating and non-migrating components across the range of diagnostics considered. Here, we place these results in the context of other modelling and observational studies of migrating and non-migrating tides and discuss the similarities and differences observed. Note that there are a large number of diagnostics which could be considered for such multi-dimensional data. Thus we must naturally restrict the discussion to a limited subsection of the data, but one which is representative of the phenomena observed in the ExUM. For observational data, we use both meteor radar data and satellite observations. The zonal and meridional wind measurements used are from a High Resolution Doppler Imager (HRDI) aboard the Upper Atmosphere Research Satellite (UARS) as well as a Doppler Imager (TIDI) aboard the NASA Thermosphere, Ionosphere, Mesosphere, Energetics and Dynamics (TIMED) explorer. Temperature measurements used are from the Sounding of the Atmosphere using Broadband Emission Radiometry (SABER) also aboard the TIMED explorer.

We consider the studies of Miyoshi et al. (2017), who used an atmosphere-ionosphere coupled model to investigate non-migrating atmospheric tides; Hagan and Forbes (2002), who used the linear mechanistic Global Scale Wave Model (GSWM) to investigate migrating and non-migrating tides in the MLT; Oberheide et al. (2011), who presented results from the Climatological Tidal Model of the Thermosphere (CTMT) from 80 - 400 km; Hibbins et al. (2019), who made observations using meteor radar wind data from the Super Dual Auroral Radar Network (SuperDARN) in the Northern Hemisphere, at around 60°N and at around 95 km; Chang et al. (2012), who compared ground-based observations of equinox diurnal tide wind fields from the first CAWSES Global Tidal Campaign with results from five commonly used models; Pokhotelov et al. (2018),

who compared meteor-radar observations made in Germany and Norway to the Kühlungsborn Mechanistic Circulation Model (KMCM); Dempsey et al. (2021), who compared meteor-radar observations at Rothera to the Whole Atmosphere Community Climate Model (WACCM) and the Extended Canadian Middle Atmosphere Model (eCMAM); Ortland and Alexander (2006), who compared observations of the diurnal tide from TIDI and UARS winds and SABER temperatures against a linear mechanistic tide model; Iimura et al. (2010), who provided an assessment of non-migrating semidiurnal tides present in TIDI wind measurements; Oberheide et al. (2006, 2007), who also examined non-migrating diurnal and semidiurnal tides in TIDI wind measurements; Wu et al. (2008a,b, 2011), who examined migrating and non-migrating diurnal and semidiurnal tides in TIDI wind measurements; Angelats i Coll and Forbes (2002), who examined both migrating and non-migrating semidiurnal tides in UARS meridional winds; Huang and Reber (2004), who examined both migrating and non-migrating diurnal and semidiurnal tides in UARS wind measurments; Zhang et al. (2006) & Forbes et al. (2008), who presented both migrating and non-migrating diurnal and semidiurnal tides in SABER temperature measurements; Li et al. (2015), who presented DE3 and SE2 tidal components from SABER temperature measurements; and Dhadly et al. (2018), who presented short-term DW1 and SW2 amplitudes from TIDI, as well as from the Navy Global Environmental Model-High Altitude version (NAVGEM-HA).

## 4.1   Non-Migrating components

We focus first on the non-migrating components produced by the ExUM and discuss these in the context of other studies of non-migrating components in the MLT.

### 4.1.1   DE3

Given the importance of DE3 in producing the wavenumber 4 structures observed in low-latitude total electron content in the ionosphere (Forbes et al., 2008), we first focus on this non-migrating component. We shall summarise the results observed in previous modelling and observational studies, and then compare with the results from the ExUM.

Miyoshi et al. (2017) considered the temperature field, and found that DE3 was the largest of all non-migrating tidal components in the MLT (peaking around 17 K amplitude at 110 km at the equator)(however at 80 km a maximal amplitude of 3 K is observed at 20S and 20N). Hagan and Forbes (2002) obtained a DE3 component of 30 K amplitude at 115 km compared to a 17 K observed amplitude. Oberheide et al. (2011) observed, in September at 100 km, a zonal wind DE3 with maximal amplitude at the equator of around 18 - 20 $\mathrm{ms}^{-1}$, no meridional wind DE3 component (DE3 is a Kelvin (equatorially-trapped) wave, and hence must have no meridional wind component), and a temperature DE3 component with maximal amplitude around the equator of around 9 K. Finally, the zonal wind field was also investigated at 90 km. The non-migrating components vanish on moving down to 90 km, i.e. the DE3 component observed previously disappears.

Considering the ExUM fields at 95 km, the DE3 component has a maximal amplitude of around 5 K in the temperature field, 17 $\mathrm{ms}^{-1}$ in the zonal wind field and 15 $\mathrm{ms}^{-1}$ in the meridional wind field. It would also be informative to consider the DE3 component produced in the ExUM at different model heights. We therefore plot this in Figure 15.

In the temperature field, we see a distinct increase in the amplitude of the DE3 component with increasing height. We see peak value of around 10 K at 107 km and a value of 4 - 5 K in September at 100 km. Whilst it is not inconceivable that the DE3

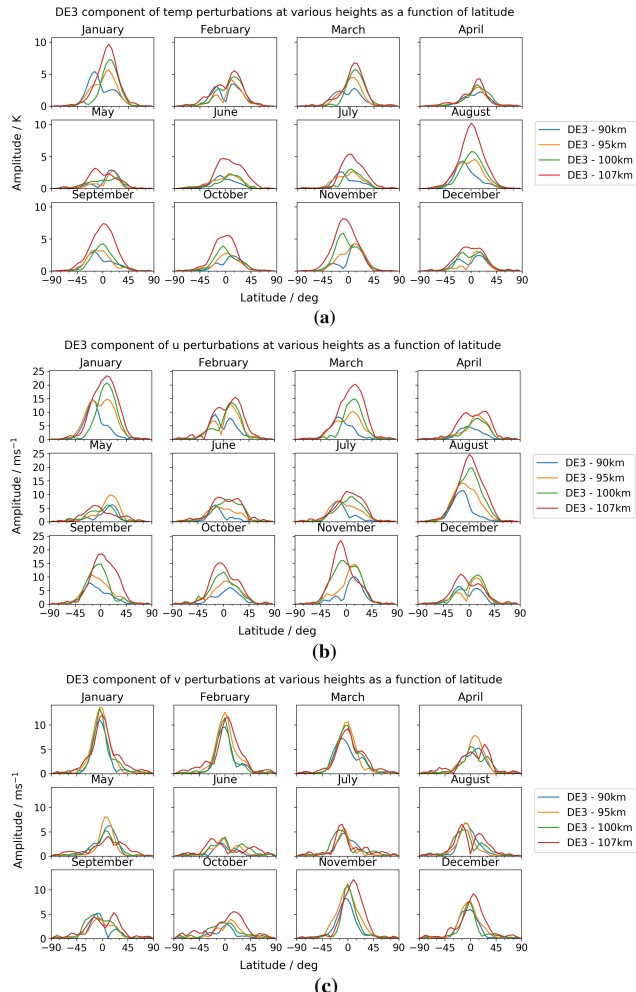

**Figure 15.** Latitude-amplitude plot of DE3 tidal amplitudes at various heights across the year for **(a)** temperature, **(b)** zonal ($u$) winds and **(c)** meridional ($v$) winds.

component could have maximal amplitudes of 17 K at 115 km, this component appears to be slightly underestimated in the modelled temperature field. In the zonal wind field, we also observe a distinct increase of the DE3 amplitude with increasing height. It reaches amplitudes of around 16 ms$^{-1}$ at 100 km in September, but has values up to around 20 ms$^{-1}$ in other months. These values are comparable to those seen in CTMT. Unlike CTMT however, the DE3 component is generally smaller at 90 km, but certainly does not disappear at this altitude. Finally, the meridional wind field is, in contrast to CTMT, non-zero at 100 km. It does not appear to greatly increase with increasing height, and actually peaks with amplitudes around 15 ms$^{-1}$ in January at 95 km.

Finally, we focus on a more detailed comparison with observational results and how they compare with the ExUM DE3 tidal fields produced.

In temperature, the satellite observations come from SABER. Forbes et al. (2008) observed DE3 amplitudes at 95 km in August/September of 6 - 8 K at around 10°S and decaying either side of this latitude. Maximal values of 12 K are observed at the same latitude at 105 - 110 km. There is a transition from a two peak structure at lower altitudes (76 km) to a single peak structure at higher altitudes (116 km) with this single peak structure having maximal values in August/September, with near zero values over equatorial winter. Zhang et al. (2006) and Li et al. (2015) echo these results. In comparison to the fields produced by the ExUM, the latitudinal structure is well captured with a transition from a two peak to single peak structure apparent with increasing altitude. The peak magnitudes are also fairly similar for the heights considered, although the ExUM perhaps slightly underestimates the DE3 component at the upper heights of the model. However it is the seasonal dependence that is the major discrepancy. A peak value is seen in August, but the peak persists for months such as November and January where small or zero values are observed in the SABER measurements.

The zonal wind is provided by satellite observations from TIDI and UARS. Oberheide et al. (2006) observed DE3 amplitudes at 95 km with maximal amplitudes of 14 $\mathrm{ms}^{-1}$ in August/September at (or just south of) the equator, growing to 18 $\mathrm{ms}^{-1}$ at around 100 - 105 km. There is a pronounced period of maximal amplitudes from July to September occurring with a single peak structure, with minimal amplitudes in December/January/February and May/June, often with a two peak structure. Wu et al. (2008b) echo these results, but with slightly larger values of 16 $\mathrm{ms}^{-1}$ at 95 km growing to around 25 $\mathrm{ms}^{-1}$ at around 105 km. Huang and Reber (2004) also echo these results, but with larger values again of 20 $\mathrm{ms}^{-1}$ at 95 km. In comparison to the fields produced by the ExUM, the peak values in August are more in keeping with Huang and Reber (2004), but are generally similar to the other studies considered. The increase in amplitude with increasing height and the latitudinal structure are also reproduced, with a single peak structure seen in months with large amplitudes, and a two peak structure often seen in months with smaller amplitudes. However, once more the seasonal dependence is the major discrepancy. Larger values are seen in August/September, with smaller values in May and June, but large values are seen in January - March where much smaller values are observed in TIDI and UARS measurements.

The meridional wind field is also provided by satellite observations from TIDI and UARS. Oberheide et al. (2006) observed DE3 amplitudes at 95 km with maximal amplitudes of 10 $\mathrm{ms}^{-1}$ in January/February at the equator, which do not grow further with increasing height. Maximal amplitudes are generally in November, December and January to April, with smaller amplitudes for the rest of the year, and with a year round single peak structure. Wu et al. (2008b) and Huang and Reber (2004) both echo these results. In comparison to the fields produced by the ExUM, we see that the ExUM DE3 meridional component has amplitudes which are slightly larger than those observed, peaking around 15 $\mathrm{ms}^{-1}$. However the latitudinal single peak structure, the seasonal structure and lack of significant increase with increasing height are all features which are also produced by the modelled meridional wind.

Ultimately then, whilst some differences do exist between the ExUM and other models and observational studies, it is notable that the DE3 component is of significant magnitude for all diagnostics considered, and is in fact one of the larger components of the motion field.

### 4.1.2 Other non-migrating components

We move our attention to other non-migrating components found in the studies introduced above.

Miyoshi et al. (2017) found other components of note in the diurnal non-migrating tides are DE2, DW2 and D0 with amplitudes of around 7 K in the 90 - 100 km region. Hagan and Forbes (2002) found that DE3 also generates DW5, SW6 and SE2 via zonal wavenumber 4 interactions and DW2, D0, SW3 and SW1 via zonal wavenumber 1 interactions. DW2 was around 5 K which was comparable with observed values. The study of Oberheide et al. (2011) using the CTMT (introduced

above) in September at 100 km observed some spread from DW1 to D0 and DW2 in the zonal wind, with amplitudes around 10 $ms^{-1}$. The meridional wind also has D0 and DW2 components with amplitudes around 10 $ms^{-1}$. Finally, the temperature field sees amplitudes spread from the DW1 to the DW2 component with amplitudes around 6 - 7 K.

Considering the ExUM temperature field at 95 km, DW2 is the largest of all non-migrating components, peaking around 11 K at the equator. DE2 and DW3 reach maximal amplitudes around 5 K, whilst other components remain below 5 K. The

720 ExUM is consistent with the studies considered in that it reproduces a large DW2 component, with magnitudes of around 5 K in September at 95 km consistent with values observed in CTMT at 100 km. The DE2 magnitude is also similar to that observed by Miyoshi et al. (2017), but we do not see magnitudes above 5 K in the D0 or DW5 components. In the wind fields at 95 km in September, DW2 peaks around 20 $ms^{-1}$, larger than that observed in CTMT, whereas D0 has peak amplitudes around 5 $ms^{-1}$, smaller than that seen in CTMT.

Miyoshi et al. (2017) observed that SW3 was the largest of all the semidiurnal non-migrating components in MLT (around 8 K at 110 km). Other components of note are SE2 and SW1 with amplitudes of around 6 K at 110 km. Hibbins et al. (2019) found that, in general, the semidiurnal tide was dominated by the SW2 (migrating) component with smaller contributions from SW1 and SW3 especially around the equinoxes. They found that the semidiurnal components maximised in the autumn equinox, with a secondary wintertime maximum. Iimura et al. (2010) demonstrated that a non-migrating SW1 is clearly present

in the MLT horizontal winds in the Northern Hemisphere, maximizing around 60°N in late spring/early summer. In addition, an SW3 and weaker S0 component were also evident in the lower thermosphere. The study of Oberheide et al. (2011) using the CTMT (introduced above) in September at 100 km observed some spread of the SW2 component in the zonal wind into the SW1 and SW3 components, with maximal amplitudes around 10 - 14 $ms^{-1}$. These components disappeared at 90 km. There are no notable non-migrating semidiurnal components in the meridional wind or temperature fields.

Considering the ExUM temperature field at 95 km, SE2, SW1, SW3 and SW4 all have similar magnitudes of around 3 - 4 K, with the peak SW3 values occurring at the equinoxes. These values are not inconsistent with the values observed by Miyoshi et al. (2017) at 110 km. In the zonal wind field at 95 km, SE2, SW1, SW3 and SW4 are again the larger components with values around 10 $ms^{-1}$. These larger semidiurnal components are in general similar to the observations of Hibbins et al. (2019) and Iimura et al. (2010). The seasonal dependence is less obvious, but at around 60°N, the SW1 component appears to be at its

largest in April. The SW1 and SW3 components in the ExUM are consistent with those observed by Oberheide et al. (2011). However, we also observe significant semidiurnal non-migrating components in the meridional wind which are not seen in their study, with values around 15 $ms^{-1}$.

Finally, we focus on a more detailed comparison with observational results and how they compare with the other ExUM non-migrating tidal fields produced. We generally look at DW2 and SW1 to limit the discussion.

Forbes et al. (2008) observed DW2 tidal amplitudes in temperature at 96 km peaking at 7 K in a single peak at the equator around November/December. The tide persists from September to February and is near zero for the rest of the year. In comparison, the ExUM has slightly larger peak amplitudes of 11 K at 95 km in December, and in general a three peak structure centred on the equator is evident (akin to the migrating tide) rather than a single peak at the equator. The seasonal dependence is generally well replicated, but with non-zero values persisting for much of the year.

Oberheide et al. (2006) observed DW2 tidal amplitudes in zonal wind at 95 km peaking at 12 $\text{ms}^{-1}$ in September at 25 - 30°S, generally occurring with a two peak structure either side of the equator. Larger magnitudes are observed from September to February with smaller amplitudes for the rest of the year. Wu et al. (2008b) includes higher latitudes, and observed larger peak amplitudes of up to 20 $\text{ms}^{-1}$ which occur in April/July at around 50°S with the smaller two peak structure replicated in some months. Huang and Reber (2004) also observe larger peak amplitudes of 16 $\text{ms}^{-1}$ occurring in July/September at around 40°S. In comparison, the ExUM DW2 tide peaks with amplitudes around 20 $\text{ms}^{-1}$ at 95 km in November/December at 25 - 30°S and 25 - 30°N. Thus the two peak structure is replicated in latitude but with larger amplitudes than those observed. As well as this, the modelled DW2 does not have larger amplitudes at higher latitudes. The seasonal dependence is also not well replicated, with small amplitudes in April and July. However larger amplitudes through the latter part of the year are generally seen. Looking at the SW1 zonal tide at 95 km, Wu et al. (2011) observed maximal amplitudes around 18 $\text{ms}^{-1}$ in November/December/January at 70 - 90°S. Values of around 9 $\text{ms}^{-1}$ persist for much of the rest of the year at high southerly latitudes. In comparison, the ExUM also has an SW2 which is large at high southerly latitudes, but with maximal values of 10 - 15 $\text{ms}^{-1}$. The seasonal dependence is also not replicated with the largest values occurring in March and August.

Oberheide et al. (2006) observed DW2 tidal amplitudes in meridional wind at 95 km peaking at 18 $\text{ms}^{-1}$ in February and September at 20 - 25°S, occurring with a two peak structure either side of the equator as in the zonal tide. The meridional tide also has larger magnitudes from September to February with smaller amplitudes for the rest of the year. Huang and Reber (2004) echo these results, as do Wu et al. (2008b), but with larger maximal amplitudes of around 25 $\text{ms}^{-1}$ observed. In comparison, the ExUM DW2 tide has a much larger peak amplitude of around 40 $\text{ms}^{-1}$ in December. The seasonal dependence is reasonably well replicated with the biggest discrepancy being the large values in May which are not seen to this extent in observations. The two peak latitudinal structure is well replicated. Looking at the SW1 meridional tide at 95 km, Oberheide et al. (2007) observes peak amplitudes of around 8 $\text{ms}^{-1}$ in January/February at around 45°S. Huang and Reber (2004) observed similarly large values in February at around 40°S. Wu et al. (2011) and Angelats i Coll and Forbes (2002) include higher latitudes. Wu et al. (2011) observe peak amplitudes of 15 - 18 $\text{ms}^{-1}$ in December at 70 - 90°S. Generally largest amplitudes are seen in October to January at high southern latitudes, with amplitudes up to 9 $\text{ms}^{-1}$ observed in February to September at high northern latitudes. Angelats i Coll and Forbes (2002) observes similar latitudinal and seasonal structure, but with smaller peak southern latitude amplitudes of around 10 $\text{ms}^{-1}$. In comparison, the ExUM has a similar peak magnitude of around 16 $\text{ms}^{-1}$ at high southern latitudes, but this is observed in August rather than December and thus the seasonal dependence is not well replicated. Amplitudes of around 9 $\text{ms}^{-1}$ are seen at high northern latitudes in April and May which are consistent with observations.

The ExUM results here reinforce the conclusions of the studies above, that there are significant amplitudes present across several non-migrating components. Generally latitudinal structure is well replicated but it is the seasonal structure which is the major discrepancy between the modelled non-migrating tides and observations.

## 4.2 Migrating components

We now focus on the migrating tidal components produced by the ExUM and discuss these results in the context of other studies of migrating components in the MLT.

Chang et al. (2012) observed that the models resolved the expected bimodal structure with tropical peaks associated with the diurnal migrating tide. Zonal wind amplitudes at 90 km ranged from around 50 ms$^{-1}$ in GSWM and eCMAM to around 25 ms$^{-1}$ in WACCM3 and to around 10 ms$^{-1}$ in TIME-GCM. In meridional wind, amplitudes at 90 km ranged from 70 - 80 ms$^{-1}$ in GSWM and ECMAM to 30 ms$^{-1}$ in WACCM3 and to 15 ms$^{-1}$ in TIME-GCM. Peak amplitudes in WACCM3 were found to occur around 15°S and 15°N whereas in GSWM and ECMAM peaks occurred around 25°S and 25°N. At 22°N, the 95 km zonal wind diurnal amplitudes increased to 65 ms$^{-1}$ in GSWM, decreased to 40 ms$^{-1}$ in ECMAM and increased to 40 ms$^{-1}$ in WACCM3. The 95 km meridional wind diurnal amplitudes increased to 100 ms$^{-1}$ in GSWM, and remained roughly constant in ECMAM and WACCM. Day to day variability or "tidal weather" was not overly present in the models used, but radar observations showed variations in zonal wind diurnal amplitudes from around 5 ms$^{-1}$ up to as much as 40 ms$^{-1}$ over a 5 day period at latitudes around peak tidal amplitudes, and variations in meridional wind diurnal amplitudes from around 15 ms$^{-1}$ up to as much as 80 ms$^{-1}$.

In comparison with these results, the ExUM also yields a two-peaked structure in the equinox diurnal tide winds fields with peaks in the tropics. The zonal wind component of this tide at 95 km peaks at 30-35 ms$^{-1}$ in equinox conditions, whereas the meridional wind component has peak amplitudes around 50-55 ms$^{-1}$. In both cases, the peaks are observed around 20 - 30°S and 20 - 30°N. This peak location is therefore closer to GSWM and ECMAM than WACCM3. Both wind components of the diurnal amplitudes observed in the ExUM fall between those observed in GSWM and ECMAM (which are larger) and those observed by WACCM3 and TIME-GCM (which are smaller), which suggests that the tidal magnitudes produced by the ExUM are at least reasonably consistent with other high-top models. The ExUM amplitudes are taken at 95 km whereas the others are taken at 90 km so some caution must be taken in drawing too many conclusions. However, it is worth noting that the values observed at 22°N by Chang et al. (2012) at 95 km also lead to the same conclusions when comparing with the ExUM amplitudes. Finally, the short term variability in zonal wind diurnal amplitudes varies from around 15 ms$^{-1}$ up to as much as 35 ms$^{-1}$ around the September equinox, with the meridional wind varying from around 30 ms$^{-1}$ up to as much as 60 ms$^{-1}$ around the September equinox. This amount of variation was not observed in the models used, and it is encouraging that the ExUM produces short term variation which is similar in magnitude to that seen in the radar observations presented in the study.

The short term variability in meridional wind diurnal and semidiurnal amplitudes can also be compared with the results of Dhadly et al. (2018). It should be taken into account that the results presented therein use a different year and data from heights between 80 and 95 km for the DW1 component and from heights between 85 and 100 km for the SW2 component. In terms of the DW1 component, the NAVGEM-HA exhibits short term variation up to around 40 ms$^{-1}$ in February/March. The short

term variation in TIDI measurements is seen to be much greater at around $70\ \mathrm{ms}^{-1}$ over the course of March. In comparison, the ExUM exhibits short term variation of around $35\ \mathrm{ms}^{-1}$ in February and June, and of around $40\ \mathrm{ms}^{-1}$ in November – values which compare favourably with those in NAVGEM-HA but differ from TIDI observations. In terms of the SW2 component, the NAVGEM-HA exhibits short term variation of as much as $80\ \mathrm{ms}^{-1}$ in March and September. The short term variation in TIDI measurements is seen to be much greater at around $120\ \mathrm{ms}^{-1}$ in many months. In comparison, the ExUM exhibits short term variation which is much weaker, of at most $15\ \mathrm{ms}^{-1}$ in April and September.

We now consider the migrating tides produced by CTMT in the study of Oberheide et al. (2011) (introduced above) in September at 100 km. In terms of the diurnal migrating tides, the zonal wind showed a two peak DW1 structure at 30°S and 30°N, with maximal amplitude of 16 - 18 $\mathrm{ms}^{-1}$. The meridional wind showed a two peak DW1 structure at 20°S and 20°N, with maximal amplitude of 26 $\mathrm{ms}^{-1}$. The temperature field showed a three peak DW1 structure with the largest peak at the equator and two smaller peaks at 40°S and 40°N. The equatorial peak has a maximal amplitude of 15 K, whilst the smaller peaks have maximal amplitudes around 6 - 7 K. Moving to the semidiurnal migrating tide, the zonal wind showed a two peak SW2 structure at 50 - 60°S and 40°N (as well as some larger values observed between 20 - 30°S) with a maximal amplitude of 28 $\mathrm{ms}^{-1}$. The meridional wind has a four peak structure, with two larger peaks at 50 - 60°S and 40°N, and smaller peaks at 20 - 30°S and 0 - 10°N. The larger peaks have amplitude around 25 - 30 $\mathrm{ms}^{-1}$ whilst the smaller peaks have amplitudes around 20 $\mathrm{ms}^{-1}$. Finally the temperature field showed a three peak SW2 structure, with the largest peak at 20°N with smaller peaks at 40°S and 10°S. The largest peak at 20°N has 13 K amplitude, whilst the smaller peaks have amplitudes around 7 - 10 K.

We again must be cautious in drawing too many conclusions in the comparison with ExUM fields taken at 95 km, but we can at least get some idea of the broad features of the modelled tides. The location of each peak is very similar in the zonal and meridional diurnal amplitudes, with maximal amplitudes around half that seen in the ExUM in both cases. The three peak temperature structure concurs with that seen in the ExUM, with similar magnitudes observed for each of the three peaks. The location of the peaks in temperature is about 10 degrees closer to the equator in ExUM compared to CTMT. Looking at the semidiurnal migrating tides, the ExUM zonal wind maximal amplitude is around 40 $\mathrm{ms}^{-1}$ in September, around 10 $\mathrm{ms}^{-1}$ larger than that seen in CTMT. The two peak structure is replicated in the ExUM, with the third smaller peak around 15°S being similar to the larger values seen in CTMT between 20 - 30°S. The ExUM meridional wind maximal amplitude is around 20 $\mathrm{ms}^{-1}$ in September, which is slightly less than that seen in CTMT. The four peak structure is also observed in the ExUM, with peaks at similar locations. Finally, the three peak structure is also observed in the ExUM temperature field in September, with the peaks in similar locations to those seen in CTMT, but with smaller magnitudes - the northmost peak is at around 7 K with the two other peaks at around 5 - 6 K.

Ortland and Alexander (2006) placed a particular focus on tuning the Gravity Wave (GW) forcing to best reflect the diurnal tide structure, rather than to focus the tuning on matching observed mean wind and temperature structure as is standard practice. They observed diurnal meridional wind amplitudes at 95 km with similar latitudinal structure to that observed in the ExUM. The peak magnitudes are on the whole slightly larger in TIDI and their tidal model, ranging from 60 - 80 $\mathrm{ms}^{-1}$, whereas we observe values around 50 $\mathrm{ms}^{-1}$ in the ExUM in March. The conclusions when comparing the diurnal temperature amplitude at

95 km are similar. A similar latitudinal structure is observed but with peak magnitudes on the whole slightly larger in SABER and their tidal model – around 20 K – than those observed in ExUM – around 15 K – in March.

Whilst the focus of our study is not on the development of the GW parameterization (the focus is rather to provide a detailed decomposition of the migrating and non-migrating components produced by the ExUM), it is nevertheless pertinent to discuss aspects of the GW parameterization here to aid future development. Particularly in the context of studies such as Ortland and Alexander (2006) and Yiğit et al. (2021).

As noted in Sect. 2, the ExUM uses the non-orographic Ultra Simple Spectral Parameterization (USSP) of Warner and McIntyre (2001), and includes frictional heating due to gravity wave dissipation, and consequent loss of kinetic energy (see Walters et al. (2017) for more details).

Ortland and Alexander (2006) found that the inclusion of GW forcing tuned for tidal structure acted to narrow the peak location by around 5 degrees. It was noted that the overall effects of gravity wave momentum forcing is highly dependent on the chosen gravity wave parameterization and chosen source spectrum. Yiğit et al. (2021) showed that implementing a latitudinally varying GW source spectrum can have a significant impact on middle atmosphere circulation, which can therefore have an important effect on the diurnal tides.

It has also been recently suggested that in-situ GW generation above the troposphere and non-primary (e.g. secondary) GW parameterization is necessary to obtain polar winter eastward winds in the MLT (Becker and Vadas, 2018, 2020) which is missing from current high-top models (e.g. Dempsey et al. (2021) in the context of WACCM and ECMAM, and Griffith et al. (2021) in the context of ExUM).

Therefore, to improve the capability of the ExUM in the MLT we recommend further studies to investigate i) the impact on the middle atmosphere mean flow structure of tuning the parameters of the USSP to produce the correct tidal structure in the MLT; ii) the appropriate latitudinal and azimuthal variation in the source spectrum of the USSP for gravity wave parameterization in the MLT; iii) the impact of such a latitudinally and azimuthally varying source spectrum on the tidal structure in the MLT as well as on the mean wind and temperatures in the middle atmosphere; iv) the impact of GW heating on tidal amplitudes in the MLT; and v) the impact of in-situ and non-primary GW generation on modelled winds and tides in the MLT.

Finally, we focus on a more detailed comparison with observational results and how they compare with the ExUM migrating tidal fields produced.

The meteor-radar observations discussed in Pokhotelov et al. (2018), show northern hemisphere semidiurnal zonal and meridional wind tidal amplitudes with values larger than $40 \, \mathrm{ms}^{-1}$, which we do not observe in the fields produced by the ExUM. However, the meteor-radar observations discussed in Dempsey et al. (2021), show southern hemisphere semidiurnal zonal and meridional wind amplitudes of 20 - $40 \, \mathrm{ms}^{-1}$, which is in keeping with the values seen in the southern hemisphere in the ExUM. It should be noted that interannual variability is currently not included in the modelled values, which could account for some of the differences observed with this study.

Satellite observations are primarily from SABER for temperature, and UARS and TIDI for winds, as introduced above. Focusing first on temperature, the study of Forbes et al. (2008) observes a DW1 at 100 km with strong three-peak structure centred on the equator with the outer peaks around 30°S and 30°N in March to May, with amplitudes around 20 K. This

becomes weaker in other parts of the year with values closer to 10 - 15 K with a pronounced low around January. SW2 at 100 km has a less clear latitudinal structure, but there tend to be maxima either side of the equator in bands from 10 - 30°N and 5 - 40°S, with the southerly peak in May to July and the northerly peak around January to March with values around 15 K. The study of Zhang et al. (2006) shows similar seasonal and latitudinal variation, but with amplitudes of 10 - 15 K for the diurnal migrating tide and 5 - 10 K for the semidiurnal migrating tide at 95 km. In comparison to the fields produced by the ExUM, we see a strikingly similar latitudinal structure in DW1 tide and a reasonably similar structure in the SW2 tide. The peak magnitudes are also similar with values of 10 - 15 K for the ExUM DW1 tide and 5 - 10 K for the ExUM SW2 tide. In the SW2 tide the seasonal variation is also similar, however it is the seasonal structure in the DW1 tide where there is the largest discrepancy. We observe peak values in the ExUM occurring in June and November with lows around August. It should be noted that the seasonal variation is significantly less pronounced, with the equatorial peak varying only between lows of 10 K and highs of 16 K.

Looking at observations of zonal winds, Wu et al. (2008a) observed peak values in the DW1 tide from TIDI at 95 km at around 30 - 40°S and 30 - 40°N with values of 60 and 40 ms$^{-1}$ respectively. These peak values tend to occur around April, but with larger values up to 40 ms$^{-1}$ seen throughout the year. Wu et al. (2011) observed peak values in the SW2 tide from TIDI at 95 km at around 50 - 70°S and 50 - 70°N with peak values around 30 - 40 ms$^{-1}$. The southern hemisphere peaks occur around April and December, whereas the northern hemisphere peaks occur around January and August. Huang and Reber (2004) observe peak values in the DW1 tide from UARS at 95 km which are slightly more equatorward, but with similar maximal amplitudes and a pronounced peak in March/April, with much lower values throughout the rest of the year. In comparison to the fields produced by the ExUM, the latitudinal structure is well reproduced, with similar, but often smaller, peak magnitudes in both the DW1 and SW2 tide. The pronounced peak in March/April is not seen in the ExUM DW1 tide, with little variation over the course of the year. We see much more variation in the SW2 tide, with the May peak in the southern hemisphere and January/August/September peak in the northern hemisphere not too dissimilar to observations. It is once more the seasonal structure of DW1 that represents the largest discrepancy.

For the meridional winds, Wu et al. (2008a) observed peak values in the DW1 tide from TIDI at 95 km at around 20°S and 20°N with values around 60 - 70 ms$^{-1}$ in March and September/October. Wu et al. (2011) observed peak values in the SW2 tide from TIDI at 95 km at around 50 - 70°S and 50 - 70°N as in the zonal wind, with peak values around 40 ms$^{-1}$, and with smaller peaks also apparent at lower latitudes. We see peak values in the southern hemisphere in June and in the northern hemisphere in August/December/January. Angelats i Coll and Forbes (2002) observe similar latitudinal and seasonal structure in the SW2 tide from UARS, with slightly larger magnitudes of up to 50 ms$^{-1}$ at 95 km. Huang and Reber (2004) observe peak values in the DW1 tide from UARS at 95 km in keeping with those observed by Wu et al. (2008a). In comparison to the fields produced by the ExUM, the latitudinal and seasonal structure as well as maximal amplitudes are very similar to observations, with no major discrepancies.

In summary, across all the tides considered, the ExUM results illustrate strong amplitude variation with latitude and month across the many components considered. There are small discrepancies in the latitudinal peak location of the modelled tides, as well as small discrepancies where tidal magnitudes are over/underestimated. However the largest discrepancy compared

with observations appears to be the seasonal structure, which is only occasionally reproduced and often differs greatly from observed values. It is possible that a factor in this discrepancy in seasonal structure is the simplified radiation and chemistry implementation used – namely that the climatological temperature profile used only gives a simple approximation for monthly and latitudinal variation compared to real values, with no interannual variation; and the ozone background files used also only give a simple approximation for monthly and latitudinal variation compared to real values, also with no interannual variation. This study therefore reinforces that the details of the gravity wave parameterization and radiation/chemistry schemes are important in the MLT and this will be the focus of future model development.

## 5  Conclusions

In this study, we perform the first in-depth analysis of migrating and non-migrating components present in the new Extended Unified Model. We have improved on the implementation of the ExUM used in Griffith et al. (2021) by i) using a monthly and latitudinally varying temperature profile above $90\,\mathrm{km}$, and ii) using a vertical resolution based on atmospheric scale height so that physically important waves are captured. We investigate the instantaneous tidal perturbations and spatial wave number decomposition at two characteristic latitudes – that of Ascension Island near the equator where the diurnal wind tide dominates, and that of Rothera at polar latitudes, where the semidiurnal wind tide dominates. We characterise the latitudinal dependence of both the diurnal and semidiurnal tide, and their variability on shorter time scales at the equatorial and polar latitudes. The model thus proves to be a useful tool for investigating migrating and non-migrating components. This is particularly useful given the difficulty in obtaining measurements of non-migrating components.

Key results include:

1. The decomposition of the modelled temperature, zonal and meridional wind fields into migrating and non-migrating tides yields significant amplitudes across a rich spectrum of temporal and spatial components.

2. The ExUM produces non-migrating components of significant amplitude in the MLT. The DW2, DE3 and DW3 components are dominant in the diurnal tide and the SW1, S0 and SW3 components are dominant in the semidiurnal tide. These components include those proposed as being key agents in thermosphere-ionosphere coupling e.g. those producing the wavenumber 4 structure in TEC in the ionosphere.

3. The migrating components are in general consistent with those reported in other modelling and observational studies. The wind fields have a bimodal latitudinal structure with tropical peaks in amplitude in the case of the diurnal tide, and with an approximate bimodal structure with amplitude peaks at polar latitudes in the case of the semidiurnal tide. The temperature field latitudinal structure reveals a three peak structure centred on the equator.

4. The ExUM suggests there is significant short-term variability in the migrating and non-migrating components – this is particularly important given the great difficulty of making experimental determinations of the short-term variability of non-migrating tides.

5. There is distinct growth in the DE3 amplitude with increasing height, from 90 km up to a height of around 105 km, where the model physics is still reasonably complete. This is an important observation given the suggested impact of DE3 in driving ionospheric variability.

6. We have proposed specific future developments of the model to improve the accuracy and physical completeness of the ExUM in the MLT, with a particular focus on the parameterization of gravity waves and the development of radiation/chemistry schemes.

In summary, our results indicate the usefulness of the ExUM in modelling atmospheric migrating and non-migrating tides in the MLT and provide insight not only into further developments required for the ExUM, but for developments within the broader context of whole atmosphere modelling.

*Code availability.* The Unified Model code is provided courtesy of the UK Met Office and is subject to copyright.

*Data availability.* The model data is produced by the UK Met Office's Unified Model, copyright UK Met Office.

*Author contributions.* The experimental concept, design of methodology and interpretation of results was performed by Griffith and Mitchell. The model development, production of model datasets and non-migrating tidal analysis of the model fields was performed by Griffith. The final authorship of the manuscript and preparation of figures was also performed by Griffith.

*Competing interests.* No competing interests are present.

*Acknowledgements.* MJG and NJM are supported by a NERC GW4+ Doctoral Training Partnership studentship from the Natural Environment Research Council [NE/L002434/1] and are thankful for the collaborative support of the Met Office, UK.

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
