# Peer review of "Analysis of Migrating and Non-Migrating Tides of the Extended Unified Model in the Mesosphere and Lower Thermosphere"

_Annales Geophysicae, 2021_

## Author Comment (AC1)

**Response to Reviewers: Comments on "Analysis of Migrating and Non-Migrating Tides of the Extended Unified Model in the Mesosphere and Lower Thermosphere" by Griffith and Mitchell**

Quotes from the reviewer are in bold, and responses are indented. We first wish to thank the reviewers for the insightful and useful comments provided on the first version of the manuscript. We will make the changes requested and believe the manuscript is strengthened as a result. We have also made some minor typographical/readability changes to the text.

**Responses to Reviewer #1:**

>We disagree that this is a technical report. The goal of this paper is firstly, to report new developments and capabilities of the ExUM, and secondly to investigate non-migrating and migrating tidal fields produced by this new model and benchmark them against observations and other modelling studies. We believe this makes the work entirely suitable for publication in Annales Geophysicae. We note that the second reviewer had none of these concerns. However, we will clarify the scientific focus in the abstract and introduction and reduce its length.

>We agree however that there is bias towards comparison of modelling studies. To this end, we will add to the discussion comparing the ExUM fields with additional satellite and meteor-radar observations. We add comparison to the studies of Zhang et al., (2006), Forbes et al., (2008), Li et al., (2015) for comparison with SABER temperatures and Wu et al., (2008a, 2008b, 2011), Pokhotelov et al., (2018), Dempsey et al., (2021), Angelats i Coll & Forbes (2002), Oberheide et al., (2006, 2007), and Huang and Reber (2004) for comparison with TIDI, UARS and meteor-radar wind measurements. However, we note that a paper regarding detailed quantitative comparison is beyond the scope of this work and is fact the goal of ongoing research.

>The overlapping material is related to introducing/specifying the extended model and so will be similar by its very nature. We agree that the overlap is in some places excessive and have removed some overlap, now referring the reader to this previous work.

References:

Angelats i Coll, M. and Forbes, J.: Nonlinear interactions in the upper atmosphere: The s= 1 and s= 3 nonmigrating semidiurnal tides, Journal of Geophysical Research: Space Physics, 107, SIA–3, https://doi.org/10.1029/2001ja900179, 2002.

Dempsey, S., Hindley, N., Moffat-Griffin, T., Wright, C., Smith, A., Du, J., and Mitchell, N.: Winds and Tides of the Antarctic Mesosphere and Lower Thermosphere: One Year of Meteor-Radar Observations Over Rothera (68° S, 68° W) and Comparisons with WACCM and880eCMAM., Journal of Atmospheric and Solar-Terrestrial Physics, 212, 105 510, https://doi.org/10.1016/j.jastp.2020.105510, 2021.

Forbes, J., Zhang, X., Palo, S., Russell, J., Mertens, C., and Mlynczak, M.: Tidal variability in the ionospheric dynamo region, Journal of Geophysical Research: Space Physics, 113, https://doi.org/10.1029/2007ja012737, 2008.

Huang, F. T. and Reber, C. A.: Nonmigrating semidiurnal and diurnal tides at 95 km based on wind measurements from the High Resolution Doppler Imager on UARS, Journal of Geophysical Research: Atmospheres, 109, https://doi.org/10.1029/2003JD004442, 2004.

Li, X., Wan, W., Ren, Z., Liu, L., and Ning, B.: The variability of nonmigrating tides detected from TIMED/SABER observations, Journal of Geophysical Research: Space Physics, 120, 10–793, https://doi.org/10.1002/2015JA021577, 2015.

Oberheide, J., Wu, Q., Killeen, T., Hagan, M., and Roble, R.: Diurnal nonmigrating tides from TIMED Doppler Interferometer wind data: Monthly climatologies and seasonal variations, Journal of Geophysical Research: Space Physics, 111, https://doi.org/10.1029/2005JA011491, 2006.

Oberheide, J., Wu, Q., Killeen, T., Hagan, M., and Roble, R.: A climatology of nonmigrating semidiurnal tides from TIMED Doppler Interferometer (TIDI) wind data, Journal of atmospheric and solar-terrestrial physics, 69, 2203–2218, https://doi.org/10.1016/j.jastp.2007.05.010,2007.

Pokhotelov, D., Becker, E., Stober, G., and Chau, J. L.: Seasonal variability of atmospheric tides in the mesosphere and lower thermosphere:1015meteor radar data and simulations, Annales Geophysicae, 36, 825–830, https://doi.org/10.5194/angeo-36-825-2018, 2018.

Wu, Q., Ortland, D., Killeen, T., Roble, R., Hagan, M., Liu, H.-L., Solomon, S., Xu, J., Skinner, W., and Niciejewski, R.: Global distribution and interannual variations of mesospheric and lower thermospheric neutral wind diurnal tide: 1. Migrating tide, Journal of Geophysical Research: Space Physics, 113, https://doi.org/10.1029/2007JA012542, 2008a.

Wu, Q., Ortland, D., Killeen, T., Roble, R., Hagan, M., Liu, H.-L., Solomon, S., Xu, J., Skinner, W., and Niciejewski, R.: Global distribution and interannual variations of mesospheric and lower thermospheric neutral wind diurnal tide: 2. Nonmigrating tide, Journal of Geophysical Research: Space Physics, 113, https://doi.org/10.1029/2007JA012543, 2008b.

Wu, Q., Ortland, D., Solomon, S., Skinner, W., and Niciejewski, R.: Global distribution, seasonal, and inter-annual variations of mesospheric semidiurnal tide observed by TIMED TIDI, Journal of atmospheric and solar-terrestrial physics, 73, 2482–2502, https://doi.org/10.1016/j.jastp.2011.08.007, 2011.

Zhang, X., Forbes, J. M., Hagan, M. E., Russell III, J. M., Palo, S. E., Mertens, C. J., and Mlynczak, M. G.: Monthly tidal temperatures 20–120 km from TIMED/SABER, Journal of Geophysical Research: Space Physics, 111, https://doi.org/10.1029/2005JA011504, 2006.

**Responses to Reviewer #2:**

**Comments:**

**Lines 147 – 168: These paragraphs to a certain degree belong to the model description. Page 6: Are these footnotes necessary? It is textbook knowledge.**

Agreed. The introductory paragraph on GW parameterization is left in the introduction and those specific to the ExUM are moved into the model description section. We agree and have removed the footnotes accordingly.

**Line 211: There has been rocket observations of mesopause temperatures lower than in CIRA (e.g. Lübken et al., 2004). Would you like to comment on that?**

Thank you for this comment. At this early stage, we use CIRA as a nudging profile for temperature as it provides a reasonable year-round temperature climatology for the MLT. We are aware that there are discrepancies between CIRA and observations such as those performed by Lübken et al., 2004, but more sophisticated temperature mechanisms and schemes will be added in the future to replace this nudging scheme and it is only intended as a reasonable estimate so that we can get a first look at the nature of the non-migrating tides in this newly extended model.

**Line 220, Figure 2: Some authors have reported a double mesopause (e.g. Yu and She, 1995). Would you like to comment on that?**

As above.

**Line 268, Figures 6: The summer wind reversal at middle latitudes seems to be a bit too low. Would you like to comment on this?**

Biases are to be expected at this early stage of model development. The objective of this paper is to test the newly extended model to discern whether the migrating and non-migrating tides produced are qualitatively reasonable and quantitively similar to other studies. These more detailed biases will be addressed in future work to improve the ExUM, where more physically realistic schemes (such as an MLT chemistry scheme) will be introduced. We will add a comment in the paper acknowledging that biases (such as you have mentioned) exist in this new extension of the model.

**Section 2: Could you add a brief description of latent heat release in the model?**

A brief sentence is added directing the reader to the more thorough overview presented in Walters et al. (2017).

**Lines 360 pp, summary of section 3.2: To what degree are these values supported by observations? This is the topic of another paper, but a brief discussion would be helpful here (also for the summary of 68°S results)**

This is a good suggestion. A brief discussion has been added following each summary, comparing the model values to observations.

**Line 362: Add here that this is at high altitudes**

This has been added to the text.

**Figure 13: Observed SDT amplitudes at NH middle latitudes have been found to be > 40 m/s (e.g. Pokhotelov et al., 2018). This seems to be more than the modeled ones. There is a SH/NH difference, and do the SH amplitudes agree better with the observations?**

Agreed, thank you for making us aware of this. The meteor radar results from Pokhotelov et al., (2018) are larger than the values we observe in the NH. Interannual variability is another source of variability between the observations and modelled values, so it is important to have this in mind when comparing the fields. The SH amplitudes seem to agree better with observations, for example those observed by Dempsey at al., (2021) at Rothera are between 20 and 40 m/s in keeping with the

values given by the ExUM. We will add comment on this in the discussion, noting that interannual variability will also play a role in the agreement of the model and observations.

---

## Referee Report (RR1)

Comments on "Analysis of Migrating and Non-Migrating Tides of the Extended Unified Model in the Mesosphere and Lower Thermosphere" by Griffith and Mitchell (Manuscript # angeo-2021-21)

This modeling study analyzes and validates the amplitude of diurnal and semidiurnal atmospheric tidal perturbations in the Extended Unified Model (ExUM). The authors find that the temporal and spatial distribution of diurnal and semidiurnal migrating and non-migrating tidal amplitudes in the ExUM compare favorably to other whole atmosphere and upper atmosphere models, as well as satellite observations on seasonal time scales. This paper also examines day-to-day variations in tidal amplitudes and finds that they can be of the same magnitude as the seasonally-derived amplitudes. The manuscript is well-written, well-organized, and suitable for publication after some major revisions.

General Comment(s):
1.) While the model results presented in Figures 1, 2, 7, 8, 10, 12, 13, 14, 15 are important, they are extremely difficult to see and are too small. I would suggest the authors make the axis labels slightly bigger and in bold type font, while also making the plots larger. This would make them much easier to read.
2.) The word "tidal modes" or "modes" is used incorrectly throughout the manuscript. Typically, "tidal modes" or "modes" in this context refers to an individual tidal components "Hough mode" or latitudinal structure. Each "Hough mode" is represented by an accompanying latitudinal function, referred to as a Hough function, or the eigen functions of Laplace's tidal equation. The latitudinal structure of any individual tidal component (e.g., DW1, SW2, or DE3) is determined by the superposition of all the different Hough modes. I would strongly suggest changing "tidal modes" or "modes" to **tidal components** or **component(s)** throughout the entire manuscript. Please see a tidal review by Forbes [1995][1] for more details.
3.) There is very little discussion throughout the manuscript about the phases of the different tidal components produced in the model. The manuscript would be greatly enhanced if there were some phase comparisons between the tidal phases produced in ExUM and other models, as well as observations.
4.) While the authors discuss how one major source of tidal dissipation is handled (ion drag, which is not included since the model top is at 120 km), there is little to no discussion about how the other major tidal dissipation processes including eddy and molecular diffusion are handled in the model. Were these discussed in previous papers? If so, one or two sentences of how these types are handled would suffice. If not there needs to be some discussion about how these things are handled, parameterized, in the dynamical core in the ExUM. A follow onto this question would be how are the specific heats handled in ExUM? Are they height varying? Please elaborate on this as well.
5.) On L165-205 there is extensive discussion on how the background temperature profile is nudged towards climatology. How robust are the simulated tidal amplitudes

[1] Forbes, J.M. (1995). Tidal and Planetary Waves. In The Upper Mesosphere and Lower Thermosphere: A Review of Experiment and Theory (eds R.M. Johnson and T.L. Killeen). https://doi.org/10.1029/GM087p0067

and phases to this background temperature profile? For example, *Jones et al.* [2018][2] showed that tidal results in the NCAR TIME-GCM were very sensitive to changes in the how the model fields were constrained. The authors should comment on how tidal results shown in Section 3 and 4 might depend on this choice of background temperature profile?

6.) L240: States " we only show results from a single simulation …" What does this mean? The simulations performed as part of this study should be clearly outlined in this work, so that independent reproducibility of results presented herein would be possible.

7.) Daily tidal amplitudes simulated in ExUM could be validated against NAVGEM-HA and TIMED/TIDI calculated by Dhadly et al. [2018][3].

Minor Comment(s):

1.) Why was the month of January chosen for a number of the plots? This was not stated in the manuscript. I do not have any issue with it, just would like to know the authors' rationale behind this. The readers might be interested in this type of information.

2.) Several times throughout the manuscript the Global Scale Wave Model (GSWM) is incorrectly referred to as the Global Wave Scale Model (GWSM). Please correct this in all places throughout the manuscript.

3.) When referring to atmospheric tidal and ionospheric coupling, I believe the authors have neglected some pretty influential pieces of work. While, I understand, it is not necessary to cite every paper I would suggest the authors cite the following work by Immel et al. [2006].[4] There are number of other works that could be cited as well, but please cite at least Immet et al. [2006].

4.) L26: Strike *etc.*

5.) L27: Strike *and tides*.

6.) L63: Replace *have been suggested to result from* to **are in part driven by**.

7.) L77: Strike *the*.

8.) L81: *Nonmigrating* should be non-migrating.

9.) L124: Replace *ask* with **seek to answer**.

10.) L128: Replace *can be suggested to* **could**.

11.) L184: What altitude is this *z* referring to? Geopotential height, geometric height, etc. Please state what the variable *z* is.

12.) L194: Strike the first *mean* and replace it with **and**. (This occurs in other places in the manuscript when referring to *zonal mean monthly mean*. Please replace this throughout the manuscript.)

13.) L210: It is not clear why WACCM-X scale heights are used? What are the purpose of these? Is this self-consistent with the use of the CIRA climatological

[2] Jones, M. Jr., Drob, D. P., Siskind, D. E., McCormack, J. P., Maute, A., McDonald, S. E., & Dymond, K. F. (2018). Evaluating different techniques for constraining lower atmospheric variability in an upper atmosphere general circulation model: A case study during the 2010 sudden stratospheric warming. *Journal of Advances in Modeling Earth Systems*, 10, 3076– 3102. https://doi.org/10.1029/2018MS001440

[3] Dhadly, M. S., Emmert, J. T., Drob, D. P., McCormack, J. P., & Niciejewski, R.(2018), Short-term and interannual variations of migrating diurnal and semidiurnal tides in the mesosphere and lower thermosphere. *Journal of Geophysical Research: Space Physics*, 123, 7106– 7123. https://doi.org/10.1029/2018JA025748

[4] Immel, T. J., Sagawa, E., England, S. L., Henderson, S. B., Hagan, M. E., Mende, S. B., Frey, H. U., Swenson, C. M., and Paxton, L. J. (2006), Control of equatorial ionospheric morphology by atmospheric tides, *Geophys. Res. Lett.*, 33, L15108, doi:10.1029/2006GL026161.

temperature model used for the background temperature? Please add additional details to address this.

14.)     Sentence on L282-283: It is not clear the dominance of the different migrating tidal components with latitude. From a theorical and observation perspective, this is what one would expect, but it is hard to see this in Figure 7. I would suggest adding some additional details to describe this switch in the latitudinal structure.

15.)     L355: Replace *non-migrating tidal components* with **tidal spectrum**.

16.)     L414-415: Please provide any details as to why DE3, DE2, DE1, DO, and DW2 magnitudes are weak in ExUM at polar latitudes?

17.)     L450: Strike *clear*.

18.)     L456: Strike *The*.

19.)     L472: Add **semidiurnal** between non-migrating and components.

20.)     L528-529: It is stated that the short term variation in DE3 is 125%. Is this relative to the 30-day running average values? If so, please state that. Seeing that the DE3 peaks at 4 K, but the short term variation is up to 5 K is puzzling. (This occurs with other waves as well, i.e., the short term variation is greater than 100%. Make sure this is clear to the reader.)

21.)     Strike *The* at the beginning of the sentences on L546 and L548.

22.)     L600: Add **are** between there and a.

23.)     L637: DE3 is a Kelvin wave (equatorially-trapped wave), please state this as the reason there is no meridional wind component.

24.)     L690: Strike *the* between DE3. (This occurs below when referring to other tidal components. Please strike those leading *the*'s as well.

25.)     L791: *Tides* should be **tide**.

---

## Referee Report (RR2)

Comments on "Analysis of Migrating and Non-Migrating Tides of the Extended Unified Model in the Mesosphere and Lower Thermosphere" by Griffith and Mitchell (Manuscript # angeo-2021-21)

This modeling study analyzes and validates the amplitude of diurnal and semidiurnal atmospheric tidal perturbations in the Extended Unified Model (ExUM). The authors find that the temporal and spatial distribution of diurnal and semidiurnal migrating and non-migrating tidal amplitudes in the ExUM compare favorably to other whole atmosphere and upper atmosphere models, as well as satellite observations on seasonal time scales. This paper also examines day-to-day variations in tidal amplitudes and finds that they can be of the same magnitude as the seasonally-derived amplitudes. The manuscript is well-written, well-organized, and suitable for publication after a few remaining points are addressed.

***Original Reviewer Comment***: *There is very little discussion throughout the manuscript about the phases of the different tidal components produced in the model. The manuscript would be greatly enhanced if there were some phase comparisons between the tidal phases produced in ExUM and other models, as well as observations.*

> **Response:** *We agree that phases are an important consideration. However, we feel that this would make the paper too long, adding many more figures. This is an interesting topic that will be addressed in future studies of the ExUM. We have added a footnote to this effect in the paper (Page 5).*
>
> Text added to manuscript: "Note that the tidal phases are also an important consideration. However, these will not be presented here to keep the paper at a reasonable length. This is an interesting topic that will be addressed in future studies of the ExUM.

**New Reviewer Comment**: I agree that keeping the paper at a reasonable length is important. However, I must push back on this boilerplate response. In my opinion, there must be some discussion about the tidal phases, especially given the lack of dissipative mechanisms (i.e., lack of eddy and molecular diffusion) in the MLT region in this version of ExUM. It is extremely important to know where the tides are no longer vertically-propagatinedg and or dissipat versus generated in-situ. I strongly suggest the authors' provide at the very least some comments and additional text on the most notable tidal components, e.g., DW1, DE3, SW2, SE3, etc phases. Simply not looking and or reporting on the tidal phases is neglecting one of the two characteristics that make a wave, a wave.

***Original Reviewer Comment***: *While the authors discuss how one major source of tidal dissipation is handled (ion drag, which is not included since the model top is at 120 km), there is little to no discussion about how the other major tidal dissipation processes including eddy and molecular diffusion are handled in the model. Were these discussed in previous papers? If so, one or two sentences of how these types are handled would suffice. If not there needs to be some discussion about how these things are handled, parameterized, in the dynamical core in the ExUM. A follow onto this question would be how are the specific heats handled in ExUM? Are they height varying? Please elaborate on this as well.*

**Response:** *We agree that additional clarification is required here, and it has been added to the text (L153 - L155). Eddy and molecular diffusion are not used in the MLT in this version of the ExUM (the paper by Griffin et al. (2018) (below) suggests this has its most dominant effect above 150 km). The specific heats are also not height varying and are just the standard values for air. This is a reasonable assumption up to the turbopause which is the primary region of interest in this paper. These are all additions which can be made to improve the model in later versions, but this paper represents a first look at tidal components from the newly extended Unified Model.*

Text added to manuscript: Other tidal dissipation processes such as eddy and molecular diffusion are not included in the MLT in this version of the ExUM (Griffin and Thuburn (2018) suggest that these effects become important at around 150 km). Furthermore, the specific heats are not height varying, which is a reasonable assumption up to the turbopause which is the primary region of interest in this study.

**New Reviewer Comment:** Eddy and molecular diffusion are certainly important below 150 km, especially eddy diffusion, which is dominant between 90-105 km. Specifically the turbopause is defined as where the time scale of molecular diffusion becomes shorter than that of eddy diffusion (Schunk and Nagy, 2009). This has been known and shown for some time in the MLT and thermospheric communities. If you say molecular diffusion does not become dominant up until about 150 km, that would be more correct, although I think a more accurate altitude would be between 110-120 km. Below I have provided just a few references for the authors' so that these statements can be revised. Stating they are not included in the model is fine, but then one wonders how are the tides dissipated in ExUM. Stating they are not important until 150 km is not what is generally accepted in the MLT and thermospheric communities. Please revise these statements with appropriate referencing to be more accurate and completely describe how the tides are dissipated in ExUM.

**Eddy and Molecular Diffusion/Tidal Dissipation Reference(s) (and references therein):**
Jeffrey M. Forbes, Maura E. Hagan,
Diurnal propagating tide in the presence of mean winds and dissipation : a numerical investigation,
Planetary and Space Science,
Volume 36, Issue 6,
1988,
Pages 579-590,
ISSN 0032-0633,
https://doi.org/10.1016/0032-0633(88)90027-X.
(https://www.sciencedirect.com/science/article/pii/003206338890027X)

Qian, L., Solomon, S. C., & Kane, T. J. (2009). Seasonal variation of thermospheric density and composition. Journal Of Geophysical Research-Space Physics, 114, 15 pp.
doi:10.1029/2008JA013643

Forbes, J. M., Zhang, X., Maute, A., & Hagan, M. E. (2018). Zonally symmetric oscillations of the thermosphere at planetary wave periods. Journal Of Geophysical Research: Space Physics, 123, 4110-4128. doi:10.1002/2018JA025258

Forbes, J. M. and R. A. Vincent, 1989: Effects of mean winds and dissipation on the diurnal propagating tide: An analytical approach. Planet. Space Sci., 37, 197–209.

Schunk, R., & Nagy, A. (2009). *Ionospheres: Physics, Plasma Physics, and Chemistry*. Cambridge: Cambridge University Press. Doi: 10.1017/CBO9780511635342

**Turbopause Reference(s):**

Zhao, X. R., Sheng, Z., Li, J. W., Yu, H., & Wei, K. J. (2019). Determination of the "wave turbopause" using a numerical differentiation method. *Journal of Geophysical Research: Atmospheres*, 124, 10592– 10607. https://doi.org/10.1029/2019JD030754.

---

## Author Response (AR2)

**Response to Second Round of Reviewers: Comments on "Analysis of Migrating and Non-Migrating Tides of the Extended Unified Model in the Mesosphere and Lower Thermosphere" by Griffith and Mitchell**

Quotes from the reviewer are in bold, and responses are indented. We first wish to thank the reviewer for the insightful and useful comments provided on the second version of the manuscript. We have made the changes requested and believe the manuscript is strengthened as a result.

**Responses to Reviewer #3:**
**General Comment(s):**
**1.) While the model results presented in Figures 1, 2, 7, 8, 10, 12, 13, 14, 15 are important, they are extremely difficult to see and are too small. I would suggest the authors make the axis labels slightly bigger and in bold type font, while also making the plots larger. This would make them much easier to read.**

> We agree, that in the current draft format the figure sizes are too small. The axis labels have been made as large as possible so that they do not overlap in response to the first round of reviews. The size of the figures is dictated by the journal format which has been adhered to in this draft. We will discuss what can be done to make the figures as large as possible with the Publications Production Office when the manuscript goes for typesetting and publication.

**2.) The word "tidal modes" or "modes" is used incorrectly throughout the manuscript. Typically, "tidal modes" or "modes" in this context refers to an individual tidal components "Hough mode" or latitudinal structure. Each "Hough mode" is represented by an accompanying latitudinal function, referred to as a Hough function, or the eigen functions of Laplace's tidal equation. The latitudinal structure of any individual tidal component (e.g., DW1, SW2, or DE3) is determined by the superposition of all the different Hough modes. I would strongly suggest changing "tidal modes" or "modes" to tidal components or component(s) throughout the entire manuscript. Please see a tidal review by Forbes [1995][1] for more details.**

> We agree. This has been changed in the manuscript.

**3.) There is very little discussion throughout the manuscript about the phases of the different tidal components produced in the model. The manuscript would be greatly enhanced if there were some phase comparisons between the tidal phases produced in ExUM and other models, as well as observations.**

> We agree that phases are an important consideration. However, we feel that this would make the paper too long, adding many more figures. This is an interesting topic that will be addressed in future studies of the ExUM. We have added a footnote to this effect in the paper (Page 5).

**4.) While the authors discuss how one major source of tidal dissipation is handled (ion drag, which is not included since the model top is at 120 km), there is little to no discussion about how the other major tidal dissipation processes including eddy and molecular diffusion are handled in the model. Were these discussed in previous papers? If so, one or two sentences of how these types are handled would suffice. If not there needs to be some discussion about how these things are handled, parameterized, in the dynamical core in the ExUM. A follow onto this question would be how are the specific heats handled in ExUM? Are they height varying? Please elaborate on this as well.**

> We agree that additional clarification is required here, and it has been added to the text (L153 - L155). Eddy and molecular diffusion are not used in the MLT in this version of the ExUM (the paper by Griffin et al. (2018) (below) suggests this has its most dominant effect above 150 km). The specific heats are also not height varying and are just the standard values for air. This is a reasonable assumption up to the turbopause which is the primary region of interest in this paper. These are all additions which can be made to improve the model in later versions, but this paper represents a first look at tidal components from the newly extended Unified Model.
>
> *Griffin, D. and Thuburn, J.: Numerical effects on vertical wave propagation in deep-atmosphere models, Quarterly Journal of the Royal Meteorological Society, 144, 567–580, https://doi.org/10.1002/qj.3229, 2018.*

**5.) On L165-205 there is extensive discussion on how the background temperature profile is nudged towards climatology. How robust are the simulated tidal amplitudes and phases to this background temperature profile? For example, *Jones et al.* [2018][2] showed that tidal results in the NCAR TIME-GCM were very sensitive to changes in the how the model fields were constrained. The authors should comment on how tidal results shown in Section 3 and 4 might depend on this choice of background temperature profile?**

> We agree that additional clarification is required here, and we have added a comment acknowledging that this is the case (L210 – L218):
> "There will naturally be some variation in the modelled tidal fields when this background temperature profile is varied *Jones et al. [2018]*. However, the main focus of this work is to provide a closer look at the migrating and nonmigrating components of atmospheric tides present in the newly extended model and show that they are of reasonable order of magnitude and compare reasonably with other models and with observations. A detailed analysis of the sensitivity of the tidal fields to the background temperature profile is beyond the scope of this work -- we note that the goal in future development of the ExUM is to replace this background temperature profile with appropriate radiation and chemistry schemes for the MLT. As well as this, the primary diagnostics used are zonal and monthly mean fields for climatological variations, which will be less sensitive to such variations in the background temperature profile. Nevertheless, it is worth bearing this in mind when considering the results presented here."

**6.) L240: States " we only show results from a single simulation …" What does this mean? The simulations performed as part of this study should be clearly outlined in this work, so that independent reproducibility of results presented herein would be possible.**

Apologies for the confusion. This means nothing more than the model is run several times with exactly the same setup to ensure robustness of the results (to phenomena such as computational glitches). This was performed and showed no difference in model results between simulations. We will change the wording of this in the manuscript for clarity (L257 – L258).

**7.) Daily tidal amplitudes simulated in ExUM could be validated against NAVGEM-HA and TIMED/TIDI calculated by Dhadly et al. [2018][3].**

Thank you for pointing out this useful reference. A comparison of the daily tidal amplitudes in the ExUM with those provided in *Dhadly et al. [2018]* has been added to the manuscript (L809 – L818).

**Minor Comment(s):**
**1.) Why was the month of January chosen for a number of the plots? This was not stated in the manuscript. I do not have any issue with it, just would like to know the authors' rationale behind this. The readers might be interested in this type of information.**

Thank you for bringing this to our attention. January is chosen solely for illustrative purposes, to give an initial insight into the tidal components present in the model, before showing more detailed tidal decompositions which are given for all months. A footnote has been added to this effect (Page 12).

**2.) Several times throughout the manuscript the Global Scale Wave Model (GSWM) is incorrectly referred to as the Global Wave Scale Model (GWSM). Please correct this in all places throughout the manuscript.**

This has been corrected appropriately.

**3.) When referring to atmospheric tidal and ionospheric coupling, I believe the authors have neglected some pretty influential pieces of work. While, I understand, it is not necessary to cite every paper I would suggest the authors cite the following work by Immel et al. [2006].4 There are number of other works that could be cited as well, but please cite at least Immel et al. [2006].**

The introduction was reduced in length at the request of the initial round of reviews. However, we agree that the Immel et al. [2006] paper in particular is a notable work and we have added this reference to the introduction (L34).

**4.) L26: Strike *etc.***

This has been corrected appropriately.

**5.) L27: Strike *and tides*.**

This has been corrected appropriately.

**6.) L63: Replace *have been suggested to result from* to *are in part driven by*.**

This has been corrected appropriately.

**7.) L77: Strike *the*.**

This has been corrected appropriately.

**8.) L81: *Nonmigrating* should be non-migrating.**

This has been corrected appropriately.

**9.) L124: Replace *ask* with *seek to answer*.**

This has been corrected appropriately.

**10.) L128: Replace *can be suggested to* could.**

This has been corrected appropriately.

**11.) L184: What altitude is this *z* referring to? Geopotential height, geometric height, etc. Please state what the variable z is.**

This z altitude is just a standard height coordinate, representing the vertical height above sea level. This has been added to the text (L186).

**12.) L194: Strike the first *mean* and replace it with *and*. (This occurs in other places in the manuscript when referring to *zonal mean monthly mean*. Please replace this throughout the manuscript.)**

This has been corrected appropriately.

**13.) L210: It is not clear why WACCM-X scale heights are used? What are the purpose of these? Is this self-consistent with the use of the CIRA climatological temperature model used for the background temperature? Please add additional details to address this.**

The WACCM-X scale heights were used to assess the variation in scale height under different solar conditions (solar minimum compared to solar maximum). These give a reasonable baseline on which to determine the vertical level set in the model. For the background temperature profile, we thought it more appropriate to use CIRA. Naturally, the WACCM-X temperature profile and the CIRA climatological temperatures used for the background temperature profile will exhibit some differences, but both will provide a reasonable initial implementation which can be tuned in future versions of the model. This has been added to the text (L226 – L228).

**14.) Sentence on L282-283: It is not clear the dominance of the different migrating tidal components with latitude. From a theorical and observation perspective, this is what one would expect, but it is hard to see this in Figure 7. I would suggest adding some additional details to describe this switch in the latitudinal structure.**

Agreed. Clarifying detail has been added to the text (L300 – L301).

**15.) L355: Replace *non-migrating tidal components* with *tidal spectrum*.**

This has been corrected appropriately.

**16.) L414-415: Please provide any details as to why DE3, DE2, DE1, DO, and DW2 magnitudes are weak in ExUM at polar latitudes?**

The reason for this difference is at this stage unknown. We will add a comment to this effect (L432 – L433).

**17.) L450: Strike *clear*.**

This has been corrected appropriately.

**18.) L456: Strike *The*.**

This has been corrected appropriately.

**19.) L472: Add *semidiurnal* between non-migrating and components.**

This has been corrected appropriately.

**20.) L528-529: It is stated that the short term variation in DE3 is 125%. Is this relative to the 30-day running average values? If so, please state that. Seeing that the DE3 peaks at 4 K, but the short term variation is up to 5 K is puzzling. (This occurs with other waves as well, i.e., the short term variation is greater than 100%. Make sure this is clear to the reader.)**

We agree that this is confusing – it is relative to the 30-day running average. This has been clarified in the text (L538 - L539 and L605, L611, L613).

**21.) Strike *The* at the beginning of the sentences on L546 and L548.**

This has been corrected appropriately.

**22.) L600: Add *are* between there and a.**

This has been corrected appropriately.

**23.) L637: DE3 is a Kelvin wave (equatorially-trapped wave), please state this as the reason there is no meridional wind component.**

This is an excellent point that we had not appreciated. We will add this to the manuscript and thank the reviewer for pointing out this interesting observation (L659 – L660).

**24.) L690: Strike *the* between DE3. (This occurs below when referring to other tidal components. Please strike those leading *the*'s as well.**

This has been corrected appropriately.

**25.) L791: *Tides* should be *tide*.**

This has been corrected appropriately.

[1] Forbes, J.M. (1995). Tidal and Planetary Waves. In The Upper Mesosphere and Lower Thermosphere: A Review of Experiment and Theory (eds R.M. Johnson and T.L. Killeen). https://doi.org/10.1029/GM087p0067

[2] Jones, M. Jr., Drob, D. P., Siskind, D. E., McCormack, J. P., Maute, A., McDonald, S. E., & Dymond, K. F. (2018). Evaluating different techniques for constraining lower atmospheric variability in an upper atmosphere general circulation model: A case study during the 2010 sudden stratospheric warming. *Journal of Advances in Modeling Earth Systems*, 10, 3076– 3102. https://doi.org/10.1029/2018MS001440

[3] Dhadly, M. S., Emmert, J. T., Drob, D. P., McCormack, J. P., & Niciejewski, R.(2018), Short-term and interannual variations of migrating diurnal and semidiurnal tides in the mesosphere and lower thermosphere. *Journal of Geophysical Research: Space Physics*, 123, 7106– 7123. https://doi.org/10.1029/2018JA025748

[4] Immel, T. J., Sagawa, E., England, S. L., Henderson, S. B., Hagan, M. E., Mende, S. B., Frey, H. U., Swenson, C. M., and Paxton, L. J. (2006), Control of equatorial ionospheric morphology by atmospheric tides, *Geophys. Res. Lett.*, 33, L15108, doi:10.1029/2006GL026161.

---

## Author Response (AR3)

**Response to Third Round of Reviewers: Comments on "Analysis of Migrating and Non-Migrating Tides of the Extended Unified Model in the Mesosphere and Lower Thermosphere" by Griffith and Mitchell**

Quotes from the reviewer are in bold, and responses are indented. We first wish to thank the reviewer for the useful comments provided on the third version of the manuscript. We have made some modifications to the paper accordingly and believe the manuscript is strengthened as a result.

**1) New Reviewer Comment: I agree that keeping the paper at a reasonable length is important. However, I must push back on this boilerplate response. In my opinion, there must be some discussion about the tidal phases, especially given the lack of dissipative mechanisms (i.e., lack of eddy and molecular diffusion) in the MLT region in this version of ExUM. It is extremely important to know where the tides are no longer vertically-propagating and or dissipated versus generated in-situ. I strongly suggest the authors provide at the very least some comments and additional text on the most notable tidal components, e.g., DW1, DE3, SW2, SE3, etc phases. Simply not looking and or reporting on the tidal phases is neglecting one of the two characteristics that make a wave, a wave.**

1. We believe the paper has useful scientific merit studying the amplitudes without considering tidal phases. Indeed, both initial reviewers were happy with this.
2. Including a proper treatment of phases would greatly increase the length of the paper. We stress that the paper is intended to be a short presentation of initial results from the ExUM which lays the groundwork for future in depth studies, such as tidal phases.

**2) New Reviewer Comment: Eddy and molecular diffusion are certainly important below 150 km, especially eddy diffusion, which is dominant between 90-105 km. Specifically the turbopause is defined as where the time scale of molecular diffusion becomes shorter than that of eddy diffusion (Schunk and Nagy, 2009). This has been known and shown for some time in the MLT and thermospheric communities. If you say molecular diffusion does not become dominant up until about 150 km, that would be more correct, although I think a more accurate altitude would be between 110-120 km. Below I have provided just a few references for the authors' so that these statements can be revised. Stating they are not included in the model is fine, but then one wonders how are the tides dissipated in ExUM. Stating they are not important until 150 km is not what is generally accepted in the MLT and thermospheric communities. Please revise these statements with appropriate referencing to be more accurate and completely describe how the tides are dissipated in ExUM.**

We thank the reviewer for bringing this to our attention and we have corrected the manuscript to address this (Footnote 2 on Page 5). In this initial stage of development of the model, tidal dissipation is primarily driven by an increase in the vertical damping coefficient which is a proxy for these dissipative processes in the model. The focus of future model development will be to include physical processes such as eddy and molecular diffusion in the model.